# Statistical Early Stopping for Reasoning Models

**Yangxinyu Xie** [1] **Tao Wang** [1] **Soham Mallick** [1] **Yan Sun** [2] **Georgy Noarov** [1] **Mengxin Yu** [3] **Tanwi Mallick** [4] **Edgar Dobriban** [1]

## Abstract

While LLMs have seen substantial improvement in reasoning capabilities, they also sometimes overthink, generating unnecessary reasoning steps, particularly under uncertainty, given ill-posed or ambiguous queries. We introduce statistically principled early stopping methods that monitor uncertainty signals during generation to mitigate this issue. Our first approach is parametric: it models inter-arrival times of uncertainty keywords as a renewal process and applies sequential testing for stopping. Our second approach is nonparametric and provides finite-sample guarantees on the probability of halting too early on well-posed queries. We conduct empirical evaluations on reasoning tasks across several domains and models. Our results indicate that uncertainty-aware early stopping can improve both efficiency and reliability in LLM reasoning, and we observe especially significant gains for math reasoning. The source code to reproduce our experiments is available at https://github.com/Xieya ngxinyu/reasoning_uncertainty

## 1. Introduction

Large language models (LLMs) have made remarkable progress in multi-step reasoning, yet they sometimes still struggle when faced with ill-posed or ambiguous queries, see e.g., Kirichenko et al. (2025); Ma et al. (2024), etc. Instead of abstaining or clarifying, models often attempt to provide definitive answers (Kirichenko et al., 2025; Ma et al., 2024). This tendency can undermine reliability and waste computation on answers that should not have been generated. A related failure mode is *overthinking*: producing unnecessarily long reasoning traces that do not improve

[1]University of Pennsylvania [2]New Jersey Institute of Technology [3]Washington University in St. Louis [4]Argonne National Laboratory. Correspondence to: Tao Wang <tawan@wharton.upenn.edu>.

*Proceedings of the 43rd International Conference on Machine Learning*, Seoul, South Korea. PMLR 306, 2026. Copyright 2026 by the author(s).

accuracy. Though reasoning can improve performance, empirical evidence suggests that verbose reasoning sometimes correlates with incorrect or uncertain predictions (Su et al., 2025; Fan et al., 2025), and that reasoning models may verbalize uncertainty without abstaining (Mei et al., 2025; Fan et al., 2025; Ma et al., 2024). Together, these findings point to a central challenge: reasoning models lack principled mechanisms to regulate reasoning dynamically in response to uncertainty.

Formally, consider a practitioner who deploys a reasoning model to interface with users performing a specific class of tasks, for instance, solving mathematical problems or answering scientific questions. In deployment, users may submit both well-posed queries (answerable through structured reasoning) and ill-posed or ambiguous ones that lack sufficient information to be answerable. The practitioner's objective is to construct a stopping rule that avoids wasteful overthinking on ill-posed queries while satisfying two key desiderata:

i. the rule should rarely halt correct reasoning on well-posed, answerable questions, formally, maintaining a false positive rate of at most $\alpha \in (0, 1)$;

ii. the rule should halt reasoning as often as possible on ill-posed or ambiguous queries, i.e., it should have high true positive rate, or power.

Once halted, the practitioner can use alternative strategies such as summarizing the reasoning trace to indicate uncertainty, requesting clarification from the user or abstaining from answering.

Prior work has proposed several strategies to address this challenge, falling into two broad categories: unsupervised and supervised approaches. Unsupervised methods rely on designing prompts to encourage models to express uncertainty or abstain when appropriate. For instance, Ma et al. (2024); Huang et al. (2025); Peng et al. (2025) develop prompts that explicitly instruct models to think critically or answer only when confident. While these methods demonstrate moderate empirical improvements and require no training data, they lack formal guarantees on false positive rates or stopping behavior.

Supervised methods, by contrast, attempt to learn when to stop from labeled data. One notable supervised approach (Zhang et al., 2025; Liu et al., 2025a; Wu et al., 2025) uses probing techniques that monitor the model's hidden activations to detect uncertainty signals during inference. Similarly, these methods require training data containing responses to both well-posed and ill-posed queries. These supervised approaches face several practical limitations. First, curating representative negative examples poses a significant challenge, as practitioners cannot easily anticipate the diverse ways users might phrase ill-formed or ambiguous questions. In contrast, it is straightforward to curate a collection of well-posed problems on which the model performs reliably. Secondly, and more importantly, the hidden states of a reasoning model may encode a superposition of multiple signals, making it difficult to interpret why a particular stop was triggered or to control false positive rates in an intuitive way. As a result, these methods may struggle to generalize when test-time queries differ from those seen during training, especially under distribution shifts. A separate line of work proposes early-stopping rules based on model uncertainty estimated from entropy or related logit-derived measures (Yang et al., 2025b; Yong et al., 2025). We will show empirically, however, such thresholds are also brittle under distribution shift. Both probing-based and entropy-based methods also require access to internal activations or logits, making them incompatible with proprietary LLMs exposed only through black-box APIs.

Alternatively, one may fine-tune models on reasoning trajectories from both answerable and unanswerable queries (Wang et al., 2025), though this approach demands substantial computational resources that may be inaccessible to many practitioners and we do not explore it further in this work.

Taken together, these limitations point to a broader and more fundamental challenge: First, these approaches provide little interpretability: while the signals extracted from hidden states or logits may correlate with uncertainty, they are difficult to understand or attribute, and practitioners cannot easily determine what the methods are reacting to when halting. Second, real-world deployments rarely satisfy the exchangeability assumptions implicit in many supervised or uncertainty-based methods, as test-time queries submitted by users need not resemble those seen during training or calibration. We also note that existing work rarely examines generalization beyond mathematical reasoning problems. Lastly, for practitioners interested in using proprietary LLMs, methods that require fine tuning, or access to internal activations or logits are simply not applicable.

Motivated by these observations, we propose a semi-supervised approach that combines interpretability with statistical rigor while testing generalizability across domains.

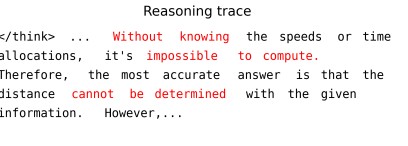 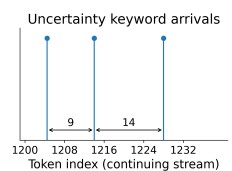

*Figure 1.* An illustration of uncertainty keyword arrival times, where inter-arrival gaps (e.g., 9 and 14 tokens) are highlighted to motivate our renewal-process–based stopping rule.

As qualitative analyses from prior work suggest that models often verbalize uncertainty in their reasoning chains (Kirichenko et al., 2025; Fan et al., 2025; Ma et al., 2024), our method proceeds in two stages. First, we use a limited seed set to identify uncertainty keywords that appear in reasoning traces. We extract these keywords by comparing reasoning traces generated in response to a set of well-posed versus ill-posed queries. This set consists of 400 pairs of well-posed and ill-posed math problems from GSM8K (Kirichenko et al., 2025), where ill-posed variants are generated via a very simple removal of information, as proposed by Kirichenko et al. (2025). These keywords, such as "impossible [to] determine" and "without [the] details," carry intuitive semantic meaning, making the stopping mechanism interpretable by design. Crucially, because our approach relies solely on the observable reasoning trace rather than internal activations or logits, it remains fully compatible with proprietary LLMs accessed only through black-box APIs. Second, we develop two statistically principled stopping rules that monitor the frequencies of these keywords during token generation, using only a calibration set of reasoning traces on well-posed problems from new, independent task domains. Critically, this calibration phase requires no knowledge of how questions might be ill-posed in the target domain and requires no negative examples. When a stopping rule is triggered, practitioners can define the model's behavior when the generation is halted, e.g., the model can return an explicit abstention response such as "I don't know."

Our methods view the occurrence of uncertainty keywords as a point process along the 'time series' of generated tokens: each time the model generates a token, we check whether it contains any uncertainty keywords. If so, we record an 'arrival' of uncertainty at that time step; see Figure 1 for an illustration. We then develop two stopping rules based on this point process.

Our first rule is inspired by renewal process theory: it models the inter-arrival times of uncertainty keywords and leverages asymptotic properties of renewal processes to construct sequential tests for stopping. Next, we introduce a nonparametric alternative based on conformal prediction that provides finite-sample guarantees, ensuring that the probability of halting too early on well-posed queries is controlled at a

pre-selected false positive rate.

Our evaluation framework focuses on the generalizability of these stopping rules: by design, we introduce distribution shifts between the calibration set (used to set thresholds) and the test set (used to evaluate performance). This allows us to interrogate the robustness of our methods when test-time queries differ from those seen during calibration. Through systematic experiments across mathematical and scientific reasoning tasks, we show that our approaches improve efficiency (cutting unnecessary tokens on ill-posed queries), while avoiding premature halts on well-posed queries and maintaining accuracy. Overall, we demonstrate that our semi-supervised, interpretable stopping rules offer a promising direction for enhancing the reliability of reasoning models in real-world deployments.

## 2. Methods

Qualitative analyses from Kirichenko et al. (2025); Fan et al. (2025); Ma et al. (2024) suggest that models often verbalize uncertainty in their reasoning traces, albeit without abstaining from answering. Inspired by these findings, we propose algorithms that leverage such uncertainty signals within the reasoning traces for early stopping. In the following, we describe our proposed framework, which consists of three stages:

1. Uncertainty Keyword Set Construction: We construct a set of uncertainty keywords using a semi-supervised procedure applied once on a small set of models and reasoning traces. This step is fully independent of model-specific calibration or test-time data used in the next two stages.

2. Stopping Rule Calibration for a Given Model: For the practitioner's target model, we use its reasoning traces responding to a set of well-posed problems to estimate uncertainty statistics and construct a model-specific stopping rule. We refer to the reasoning traces used in this step as the *calibration traces*.

3. Test-Time Application: For new user queries, whether well-posed or ill-posed, we apply the calibrated stopping rule during decoding to determine, in real time, whether to continue or halt reasoning. We refer to the reasoning traces generated at test time as *test traces*.

### 2.1. Uncertainty Keyword Set Construction

We construct an uncertainty keyword lexicon using a semi-supervised, interpretable feature selection pipeline applied to reasoning traces. Starting from 800 paired traces of well-posed and unanswerable GSM8K problems generated by four reasoning models, we train random-forest classifiers on $k$-gram bag-of-words features to discriminate between the two trace types. Uncertainty-related features are identified by intersecting high-importance $k$-grams across cross-validation folds. We then retain only interpretable uncertainty expressions and categorize them into three semantically meaningful groups-Impossibility, Speculation, and Insufficiency-via rule-based lexical filtering, with Insufficiency capturing missing-information cues most indicative of ill-posedness. The resulting uncertainty set $\mathcal{K}$ contains 102 keywords and is used by our stopping rules, while auxiliary categories capturing generic epistemic or transitional language are reserved for ablations. This procedure yields a compact, interpretable lexicon that generalizes across models and domains while requiring only a small, easily constructed seed dataset. More details of the implementation are provided in Appendix A.1.

### 2.2. Uncertainty Keywords as a Renewal Process

In this paper, we view the occurrences of uncertainty keywords in reasoning traces as arrivals in a renewal process. Let $T = (t_1, t_2, \ldots, t_L)$ denote a tokenized reasoning trace of length $L$ tokens. We define the arrival times $\{X_1, X_2, \ldots, X_k\}$ as the token positions where uncertainty keywords in $\mathcal{K}$ are detected, obtained via the greedy longest-match algorithm described in Appendix A.2. The algorithm identifies an arrival by matching the longest possible keyword phrase in $\mathcal{K}$ starting from each token position in $T$.[1] This view enables us to propose two early stopping rules based on the statistical properties of the frequency and timing of these uncertainty signals. Throughout our discussion, we assume the practitioner would like to control the false positive rate of early stopping on well-posed queries at a user-specified level $\alpha \in (0, 1)$, e.g. $\alpha = 5\%$.

In this stage, we collect a set of independent and identically distributed calibration traces $\{T^{(i)}\}_{i=1}^n$ from the practitioner's target model, where each trace $T^{(i)}$ corresponds to the model's reasoning on a well-posed query. We then use these traces to calibrate two stopping rules, described next. Notice that these calibration traces do not require any knowledge of how questions might be ill-posed in the target domain, as such enumerating ill-posed queries is a formidable challenge in practice. Later in Section 3, we stick to this setup and evaluate the generalizability of our methods under distribution shifts between calibration and test sets.

**Renewal Process Stopping**  Our first approach is a parametric rule based on renewal process theory (Grimmett and Stirzaker, 2001). This rule stops decoding when the observed arrival rate of uncertainty phrases is significantly

---

[1] The traces are preprocessed in the same way as in Section 2.1 before matching; we also mitigate over-counting highly correlated keywords by skipping the next 5 grams after each detected arrival.

higher than expected. From the calibration trace $T^{(i)}$, we extract inter-arrival times $A_j^{(i)} = X_{j+1}^{(i)} - X_j^{(i)}$, $j = 1, \dots$. Define the pooled set $\mathscr{A} = \{A_j^{(i)} : i = 1, \dots, n, \ j = 1, \dots, k_i - 1\}$ where $k_i$ is the number of arrivals in trace $i$, let $M_{\text{pool}} = |\mathscr{A}|$, then $\hat{\mu} = (1/M_{\text{pool}}) \sum_{(i,j)} A_j^{(i)}$ and $\hat{\sigma}^2 = (1/(M_{\text{pool}} - 1)) \sum_{(i,j)} (A_j^{(i)} - \hat{\mu})^2$.[2]

For a test trace, let $N_t$ be the number of arrivals up to position $t$. Renewal theory (see Grimmett and Stirzaker, 2001, Sec. 10.2, p. 417) implies that the normalized statistic $Z_t = (N_t - t/\hat{\mu}) / \sqrt{t\hat{\sigma}^2/\hat{\mu}^3}$ converges in distribution to a standard normal distribution as $t \to \infty$, if the reasoning trace corresponds to a well-posed query.[3] If $Z_t$ exceeds $z_{1-\alpha'}$, the $(1 - \alpha')$ standard normal quantile, we halt generation at step $t$. To account for repeated testing, we adjust the significance level using the Sidák correction (Šidák, 1967): $\alpha' = 1 - (1 - \alpha)^{1/\chi}$, where $\chi = \left\lfloor \frac{L_{\max}}{B} \right\rfloor$ is the maximum number of tests performed for this trace. Here, $L_{\max}$ is the maximum allowed length of the reasoning trace, and $B$ is a user-specified bin size that determines how frequently we check for stopping. Specifically, we check for stopping at token positions $t = B, 2B, 3B, \dots$.[4]

**Maxwise Conformal Stopping**   Our next approach leverages an uncertainty scoring function based on the density of uncertainty keywords in a reasoning prefix. Specifically, for a tokenized trace prefix $T[1 : \ell]$ of length $\ell > 0$, we define an uncertainty density score $u(T; \ell)$ as:

$$u(T; \ell) = \#\{\text{arrivals of keywords in } T[1 : \ell]\}/\ell. \quad (1)$$

Then, this approach computes the maximum of the uncertainty score over the traces in the following way: First, we partition traces into bins of tokens of size $B$, with boundaries $L_j = j \cdot B$, $j = 1, \dots$. For each calibration trace $T^{(i)}$ and boundary $L_j \leq |T^{(i)}|$, we compute the score $u(T^{(i)}; L_j)$ on the prefix $T^{(i)}[1 : L_j]$. We then define a global threshold of "maximal uncertainty" as follows. For each calibration

---

[2] If there are less than 2 arrivals in a calibration trace, we use the length of the trace as a single inter-arrival time, although this is an underestimation. We make this choice to avoid discarding traces with few arrivals, which could also bias the estimates downwards and increase variance.

[3] While the result applies for renewal processes with i.i.d. interarrival times, we use it as a working approximation for LM-generated traces.

[4] Technically, this correction requires independence, which does not hold in our case. Moreover, the Sidák correction is known to be conservative when tests are correlated (Holland and Copenhaver, 1987). Another practical challenge is that we cannot know $L_{\max}$ in advance, at least not precisely or making strong assumptions. We view the renewal theory as a practical approximation rather than an exact guarantee. In our experiments, we set $L_{\max}$ to be the median length of calibration traces, and we find that this method performs reasonably well in our experiments (see Section 3).

trace $T^{(i)}$, compute $M_i = \max_{j : L_j \leq |T^{(i)}|} u(T^{(i)}; L_j)$. Then, to quantify how large these scores typically get during normal reasoning, we calculate their $(1 - \alpha)(1 + 1/n)$ quantile, for some user-specified $\alpha \in (0, 1)$, i.e., let $k = \lceil (n + 1)(1 - \alpha) \rceil$ and set $\tau^{\star} = M_{(k)}$ (the $k$-th smallest value).[5] For a new query $T$ at test-time, we monitor $u(T; L_j)$, $j = 1, \dots$ and stop as soon as $u(T, L_j) > \tau^{\star}$. This controls the probability of early stopping on a well-posed query at level $\alpha$.

---

**Algorithm 1** Maxwise Conformal Stopping (Uncertainty-based)

---

1: **Input:** Calibration set $\{T^{(i)}\}_{i=1}^{n}$, bin size $B$, confidence level $\alpha \in (0, 1)$
2: **Output:** Global stopping threshold $\tau^{\star}$
3: Define bin boundaries $L_j = j \cdot B$ up to max trace length.
4: **for** each calibration trace $T^{(i)}$ **do**
5:    Compute $M_i = \max_{j : L_j \leq |T^{(i)}|} u(T^{(i)}; L_j)$.
6: **end for**
7: Set threshold $\tau^{\star} = \text{Quantile}_{(1-\alpha)(1+1/n)}\left(\{M_i\}_{i=1}^{n}\right)$.
8: **Return** $\tau^{\star}$.
9: **Prediction Phase:**
10: For a new trace $T$, after the first bin boundary $L_1$, check uncertainty at bin boundaries.
11: Stop if $u(L_j) > \tau^{\star}$ for some $j$.

---

**Proposition 2.1.** *Let $M_i = \max_{j : L_j \leq |T^{(i)}|} u(T^{(i)}; L_j)$ for calibration traces, and $M_{n+1}$ for the test trace. Define $\tau^{\star}$ as in Algorithm 1. Then*

$$\mathbb{P}\left(\exists j : u(T^{(n+1)}; L_j) > \tau^{\star}\right) = \mathbb{P}(M_{n+1} > \tau^{\star}) \leq \alpha.$$

The proof is deferred to Appendix C.

## 3. Experiments

### 3.1. Experimental Setup

We outline the models, prompts, and evaluation metrics used in our experiments below. For more details, please refer to Appendix A.3.

**Models, Instruction Prompts, and Hyperparameters**
We evaluate 12 reasoning models, spanning families such as DeepSeek (Guo et al., 2025), Qwen (Yang et al., 2025a), Nemotron (Liu et al., 2025b), MiMo (Xiaomi et al., 2025), and Skywork (He et al., 2025); the full list is in Table 6. For simplicity, we append the identical zero-shot instruction prompt after the task description and do not apply any model-specific tuning.

For efficient implementation, we compute the uncertainty scores at intervals of 250 tokens during generation, and only

---

[5] When $(1 - \alpha)(1 + 1/n) > 1$, we set the quantile to be $+\infty$

check the stopping condition at these intervals. In other words, for the Renewal stopping rule, we only compute $Z_t$ if $t$ is a multiple of 250; meanwhile, for the Maxwise stopping rule, set $B = 250$. We also conduct experiments to assess the impact of this hyperparameter. The details are provided in Appendix B.2.

**Metrics** We evaluate each method for *false positive control* and *power/efficiency*. All metrics are averaged across models and datasets within each domain.

**False Positive Rate (FPR) Control:** Defined as the probability of stopping too early on an well-posed query (i.e. the *early stopping rate on original benchmarks*). Lower values indicate stronger FPR control. For the prompting baselines, there is no explicit early stopping mechanism; instead, we measure FPR by the accuracy drop on well-posed queries compared to no intervention, which could be negative due to the stochastic nature of LLMs.

**Power / Efficiency:** The ability to stop early in ill-posed cases while not truncating useful reasoning otherwise. We measure this using: (a) *Early stopping rate on ill-posed benchmarks* (Power; higher values are better): the frequency of early stopping on ill-posed queries. For the prompting baselines, we use the abstention rate as a proxy. We use an LLM to classify whether the model abstained from answering; the detailed procedure is in Appendix A.3. (b) *Token savings on ill-posed benchmarks* (Efficiency): the percentage of tokens saved relative to the full trace length.

### 3.2. Baselines

**Prompting Baselines** We distill two prompting baselines from prior work: 1. Confidence Dampening Prompt: Following (Huang et al., 2025), append: *"Answer only if you are confident. Otherwise, say 'I am not sure.' "* to encourage abstention. 2. Critical Reasoning Prompt: Following (Ma et al., 2024), append: *"Please solve these problems with criticism. If the problem is reasonable, think step by step and put your final answer in a box. If the problem is unreasonable, highlight the issues clearly and provide a succinct explanation."* to encourage critique of unreasonable inputs while maintaining structured reasoning.

**Length-Based Baseline** Previous work suggests that reasoning models generate longer responses for ill-posed queries (Fan et al., 2025; Ma et al., 2024; Su et al., 2025). As a baseline, we therefore limit the trace length uniformly across all examples. Given calibration traces, a length threshold $\tau$ is computed as the $(1 - \alpha)(1 + 1/n)$-quantile of trace lengths $|T^{(i)}|$, $i = 1, \ldots, n$ in our calibration dataset. For new queries, generation continues until the length exceeds $\tau$, at which point token generation stops. Under exchangeability conditions, this method effectively controls

the false positive rate (FPR)—defined here as the probability of halting too early on a well-posed query (Vovk et al., 2005; Laxhammar and Falkman, 2011).[6]

**Logit-Based Uncertainty Measures** While our methods focuses on uncertainty scores derived from keyword arrivals, the Maxwise conformal framework in Algorithm 1 applies to any scalar-valued sequence of scores observed along the reasoning trajectory. To compare our keyword-based uncertainty signal against these logits-based alternatives, we substitute $u(T; \ell)$ with using (i) the DEER confidence score (Yang et al., 2025b) and (ii) a beam-search–based entropy over the answer space (Yong et al., 2025). Implementation details of these variants are provided in Appendix A.6.

**Probing-Based Baseline** We compare against the probing-based uncertainty detection method proposed by Liu et al. (2025a), which trains a linear probe on the model's hidden activations to classify whether a query is ill-posed. This is described in detail in Section 3.3.3.

### 3.3. Evaluation on Math Reasoning Tasks

First, we investigate the performance of our proposed stopping rules on math reasoning tasks. Additional experiments on scientific reasoning tasks and detailed ablation studies of our methods are provided in Appendix B. Our workflow involves three stages: (1) extracting uncertainty keywords from the training split of GSM8K; (2) calibrating the stopping rule on a separate calibration split of GSM8K to achieve the desired FPR level; (3) evaluating the calibrated stopping rule on multiple independent ill-posed math benchmarks, inclduing GSM-MC, UMWP, MiP, and MMLU (math subsets). The workflow is illustrated in Figure 2, and the datasets used are summarized in Table 7. We remark that although theoretical FPR control is guaranteed when perfect exchangeability between the calibration and test sets holds, strictly satisfying this requirement is almost never possible in practice. Therefore, rather than enforcing exact exchangeability, we instead evaluate our method on a diverse collection of math datasets, each constructed with different mechanisms for inducing ill-posed versions, to assess the robustness and practical reliability of our stopping rules. For the well-posed queries, we note that only GSM-MC contains GSM8K problems; the other three datasets are all out-of-distribution math benchmarks.

We visualize the resulting distribution shift in Figure 3. The figure reports empirical quantiles of reasoning trace lengths produced by two different families of reasoning models when responding to well-posed problems from each benchmark. As expected, GSM8K and GSM8K-MC exhibit nearly identical length distributions across all quan-

---

[6]For completeness, a proof is provided in Appendix A.4.

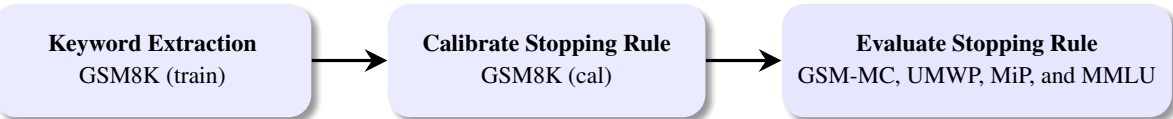

*Figure 2.* Workflow for extraction, calibration, and testing of the stopping rule.

tiles. In contrast, out-of-distribution benchmarks induce markedly different length profiles. Notably, UMWP consistently yields shorter reasoning traces, particularly for DeepSeek models, while benchmarks such as MMLU and MiP produce much longer traces with heavier upper tails. These observations indicate that reasoning length can shift in both directions under distribution shift and is strongly model-dependent.

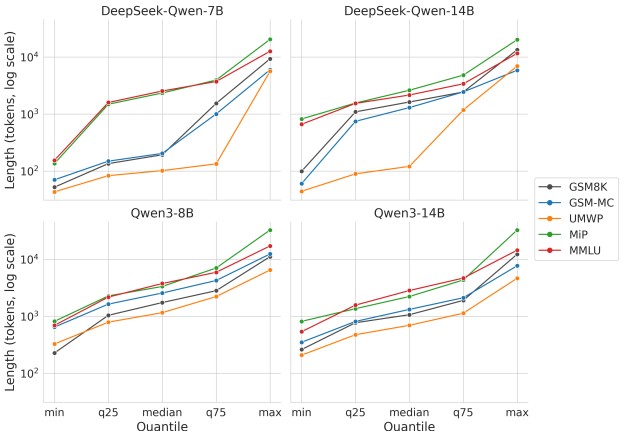

*Figure 3.* Quantiles of reasoning trace lengths (log scale) for well-posed problems across math benchmarks and two model families. Each panel corresponds to a model, and each curve shows the distribution of trace lengths on a benchmark, summarized by the minimum, 25th percentile, median, 75th percentile, and maximum. Substantial variation across datasets reveals a clear distribution shift in reasoning length, even when all queries are answerable.

As shown in Table 1, although the prompting baselines achieve good FPR control, they also exhibit almost no power in detecting ill-posed queries. Meanwhile, length-based stopping rule can be sensitive to the distribution shift between calibration and test sets, leading to poor FPR control: as noted earlier, UMWP tends to induce shorter reasoning traces while MiP and MMLU lead to longer ones. This results in under- and over-shooting of the length threshold, respectively, causing excessive false positives or missed opportunities for early stopping. Even for GSM-MC, which is in-distribution with respect to GSM8K, the length-based method only produces moderate power, stopping early on less than one-third of ill-posed queries, suggesting that length alone is not a sufficiently informative signal for early stopping.

In contrast, our proposed uncertainty-driven stopping rules (Maxwise and Renewal) consistently achieve strong FPR

control while maintaining high detection power across all math benchmarks. We observe that the Renewal stopping rule generally achieves better FPR control than Maxwise, while Maxwise attains slightly higher detection power. Similarly, we observe a higher token savings on ill-posed queries for Maxwise compared to Renewal, as shown in Table 2. Similar to the length baseline, both Renewal and Maxwise exhibit lower power on UMWP compared to other benchmarks. This is because of the shorter reasoning traces induced by UMWP, which is often below our choice of $B$ (250 tokens), limiting the opportunities for our method to apply before the reasoning ends. As reflected by the quantile statistics in Figure 12, a large fraction of UMWP responses, particularly for DeepSeek and Skywork models, terminate before reaching the first evaluation point of the stopping rule, thereby reducing detection power. We further investigate the impact of $B$ in Appendix B.2.

### 3.3.1. SOFT UPPER BOUND OF DETECTION POWER

It is important to remark that because our method is built on the semantic space of the reasoning traces, there is an inherent limit to the detection power achievable by any keyword-based approach that relies solely on the reasoning traces. If, by reading the reasoning traces, it is not possible to distinguish whether it is responding to an ill-posed or well-posed query, then no method can succeed in this task.

Therefore, to understand the results outlined in the previous section and to assess the effectiveness of our keyword-based uncertainty detection method, we establish a soft upper bound on detection power using an oracle approach. We develop the oracle by recycling the random forest classifier training procedure outlined in Section 2.1 and apply cross-validation on the test sets of datasets in Table 7. Specifically, on each evaluation fold, we report the true positive rate corresponding to a false positive rate equal to $\max\{\alpha, \widehat{\text{FPR}}\}$, where $\widehat{\text{FPR}}$ is the false positive rate induced by the stopping rule to compare against. We then average this quantity across the two folds. We use two folds because some datasets are small, with as few as 52 samples (MiP in Table 7). We use this approach to estimate the maximum achievable detection power under a supervised setting with knowledge of how ill-posed queries are constructed.[7] By comparing our method against this oracle, we can gauge

---

[7] We emphasize that our upper bound is *soft* because it relies on a learned classifier that may not be optimal.

| Stopping Rule | GSM-MC | | UMWP | | MiP | | MMLU | |
|---|---|---|---|---|---|---|---|---|
| | $\Delta^\downarrow$ Acc | Abstention | $\Delta^\downarrow$ Acc | Abstention | $\Delta^\downarrow$ Acc | Abstention | $\Delta^\downarrow$ Acc | Abstention |
| No Intervention | – | 6.75% | – | 8.95% | – | 1.12% | – | 0.50% |
| Confidence | 0.42% | 5.42% | -0.26% | 10.05% | -0.48% | 0.48% | 4.95% | 12.16% |
| Criticism | 0.67% | 5.67% | 0.26% | 9.61% | 0.32% | 0.96% | -0.50% | 15.48% |
| | FPR | Power | FPR | Power | FPR | Power | FPR | Power |
| Length | 4.50% | 32.17% | 0.62% | 8.59% | 20.67% | 61.54% | 12.91% | 39.85% |
| DEER | 5.08% | 37.25% | 2.12% | 18.86% | 16.67% | 62.34% | 20.18% | 52.63% |
| Entropy | 4.83% | 44.92% | 1.32% | 15.31% | 19.39% | 55.45% | 2.57% | 15.29% |
| **Renewal** | 3.75% | 69.75% | 2.92% | 46.78% | 1.12% | 71.15% | 0.75% | 66.42% |
| **Maxwise** | 5.42% | 70.92% | 5.52% | 48.65% | 3.21% | 73.40% | 2.63% | 75.31% |

*Table 1.* Early stopping rates across math reasoning benchmarks. "FPR" = stopping too early on well-posed queries. "Power" = stopping on ill-posed queries. Averages are taken over both all models evaluated. Confidence refers to the confidence dampening prompting baseline, and Criticism refers to the critical reasoning prompting baseline. For Confidence and Criticism, FPR is measured by accuracy drop and Power by abstention rate, which can be negative. -0.00% indicates a very small negative number rounded to two decimal places.

| Stopping Rule | GSM-MC | UMWP | MiP | MMLU |
|---|---|---|---|---|
| Renewal | 60.69% | 38.29% | 62.78% | 58.80% |
| Maxwise | 63.30% | 41.10% | 68.15% | 69.83% |

*Table 2.* Average percentage of tokens saved on ill-posed queries. Averages are taken over both all models evaluated.

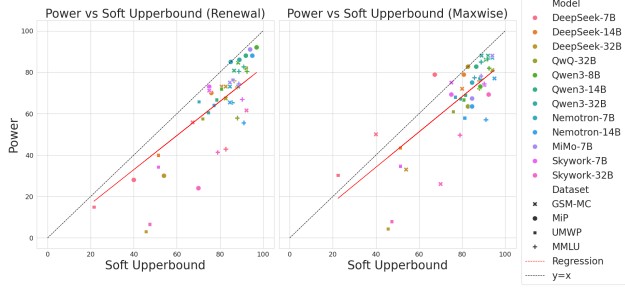

*Figure 4.* Comparison of early stopping rates between our proposed methods and the oracle upper bound across math reasoning benchmarks, where different shapes and colors correspond to different datasets and models, respectively. The regression slopes are 0.8238 and 0.8532, respectively, for Renewal and Maxwise.

how closely our unsupervised approach approximates the ideal detection capability.

As illustrated in Figure 4, our proposed uncertainty-based stopping rules achieve detection power that approaches the oracle upper bound across all math reasoning benchmarks (close to the 45-degree line). We run linear regression without intercept to quantify this relationship, finding slopes of 0.8238 and 0.8532 for Renewal and Maxwise respectively. This indicates that on average, our methods achieve approximately 82-85% of the oracle's detection power.

### 3.3.2. COMPARISON WITH DEER- AND ENTROPY-BASED STOPPING

We next compare our keyword-based Maxwise rule to logits-based early stopping methods instantiated under the same Maxwise framework. As shown in Table 1, our keyword-based Maxwise rule consistently achieves competitive detection power compared to both DEER- and entropy-based methods across all math reasoning benchmarks. More importantly, DEER and entropy exhibit substantially higher FPR on MiP and MMLU, compared to our keyword-based Maxwise rule, pointing to the greater brittleness of logits-based uncertainty signals under distribution shift in the query space. This gap is especially striking given that all three methods share the same conformal calibration and stopping logic; the only difference is the underlying uncertainty signal.

### 3.3.3. COMPARISON AGAINST PROBING METHODS

In this section, we compare our uncertainty-based stopping rules against the linear probing method proposed by Liu et al. (2025a), which uses the output of the multi-head attention before the residual connection as the input to the linear probe. Because choosing the optimal layer to probe is an art in itself, we only evaluate on four models, whose optimal layers were identified by Liu et al. (2025a). Full details of dataset construction and probe training are deferred to Appendix A.5.

During inference, we apply the trained probe to the token-level activations generated during the reasoning process. To ensure fair comparison with our uncertainty-based stopping rules, we probe the model's activations at intervals of 250 tokens. If at any point the probe predicts that the question is ill-posed with probability greater than a threshold $\tau$, we stop the generation early. We tune $\tau$ on the GSM8K calibration set: to achieve the desired FPR level $\alpha$, we first take the maximum among the predicted probabilities for each

| Model | Stopping Rule | GSM-MC | | UMWP | | MiP | | MMLU | |
|---|---|---|---|---|---|---|---|---|---|
| | | FPR | Power | FPR | Power | FPR | Power | FPR | Power |
| DeepSeek -14B | Probe | 3.00% | 64.00% | 4.39% | 56.14% | 30.77% | 88.46% | 30.83% | 97.74% |
| | PLS | 5.00% | 68.00% | 6.14% | 51.75% | 7.69% | 84.62% | 2.26% | 87.22% |
| | **Renewal** | 5.00% | 70.00% | 2.19% | 39.91% | 1.92% | 73.08% | 0.75% | 72.93% |
| | **Maxwise** | 9.00% | 72.00% | 2.63% | 43.42% | 3.85% | 78.85% | 3.01% | 75.94% |
| Qwen3 -14B | Probe | 6.00% | 88.00% | 2.63% | 67.98% | 40.38% | 94.23% | 40.60% | 95.49% |
| | PLS | 11.00% | 99.00% | 13.60% | 89.91% | 19.23% | 98.08% | 12.78% | 95.49% |
| | **Renewal** | 4.00% | 88.00% | 2.63% | 66.67% | 3.85% | 80.77% | 1.50% | 82.71% |
| | **Maxwise** | 4.00% | 88.00% | 6.58% | 68.86% | 5.77% | 82.69% | 6.02% | 86.47% |

*Table 3.* Early stopping rates comparing probing-based methods (Liu et al., 2025a) against our proposed uncertainty-based stopping rules on math reasoning benchmarks for 14B-scale models.

reasoning trace corresponding to a well-posed question, and then set $\tau$ to be the $(1 - \alpha)$-quantile of these maximum predicted probabilities.

Evaluation results for `Qwen3-14B` and `DeepSeek-14B` are reported in Table 3, with additional results for smaller models deferred to Appendix A.5. As shown in Table 3, the probing-based method can achieve both reasonable FPR control and high detection power when the distribution shift between calibration and test sets is small (e.g., GSM-MC and UMWP). However, when the distribution shift is larger (e.g., MiP, MMLU), the probing-based method suffers from significant FPR inflation, leading to a high rate of premature stopping on well-posed queries. This highlights the drawback of linear probing: the hidden activations may be comprised of the superposition of multiple kinds of information, and the probe may inadvertently latch onto spurious correlations that do not generalize well under distribution shifts, making conformal FPR control challenging.

In contrast, our proposed uncertainty-based stopping rules consistently maintain strong FPR control while achieving competitive detection power across all benchmarks, demonstrating their robustness to distribution shifts.

To further examine whether hidden activations encode multiple entangled factors of variation, we replace the linear probe with a partial least squares (PLS) regression model restricted to a single latent component. PLS explicitly seeks a one-dimensional direction that maximally covaries with the ill-posedness label, thereby isolating the dominant linear signal available for discrimination. We remark PLS is used purely as a diagnostic tool rather than a competitive classifier. As shown in Table 3, PLS substantially reduces FPR on out-of-distribution datasets MiP and MMLU compared to linear probing, supporting the superposition hypothesis that probe performance may rely on fragile combinations of entangled signals. Further discussion is deferred to Appendix A.5.

## 4. Discussion and Conclusion

We introduce two efficient mechanisms for truncating reasoning traces based on detecting uncertainty cues, with theoretical false-positive control under exchangeability assumptions. Empirically, our methods achieve reliable early stopping on ill-posed queries while maintaining low false-positive rates across diverse datasets spanning math and scientific domains. In contrast, the prompting and length-based baselines fail to produce meaningful improvements. Our results are thus complementary to previous findings (Kirichenko et al., 2025; Fan et al., 2025) that ill-posed problems inherently trigger longer reasoning. Instead, our data show that some models may not necessarily generate longer traces when problems are ill-posed, making length an unreliable stopping signal and requiring more nuanced uncertainty-based strategies. Moreover, when distribution shifts occur between calibration and test sets, length-based thresholds can become miscalibrated, leading to elevated false-positive rates or reduced power. This directly supports the motivation that models lack principled ways to regulate their own reasoning depth, and that overthinking cannot be reliably mitigated through length alone.

Across tasks, our uncertainty-driven stopping rules achieve strong false-positive control while offering substantial token savings on ill-posed queries. These improvements arise despite the methods relying solely on explicit reasoning text, without access to hidden states or further supervised training on negative examples. The resulting mechanism is therefore lightweight and compatible with black-box LLM APIs. On the other hand, because our methods operate purely on the reasoning text, their achievable power is fundamentally limited by the information about ill-posedness that is expressed in the trace. When a model's reasoning does not reflect ambiguity or missing information in a way that is distinguishable from normal deliberation, no trace-only detector can succeed. To contextualize performance, we introduce an oracle upper bound and show that our methods recover approximately 82-85% of the achievable power on math benchmarks, indicating that a large fraction of dis-

criminative signal is captured by simple keyword-arrival statistics. Finally, our approach relies on the bin size parameter. Infrequent checks improve efficiency but can reduce power when traces terminate early. Developing an adaptive checking schedule could be a future direction to balance computational cost and detection accuracy.

Our work focuses on the design of statistically principled stopping criteria, which we view as a module within a deployment pipeline. Once a stopping event is triggered, different downstream policies can be applied depending on the application context. The design of downstream policies may require a systematic study of how these strategies affect user experience. In practice, premature stopping or excessive abstention may affect user experience. Careful choice of level $\alpha$, or additional safeguards and human oversight, should be considered in deployment for high-stakes use cases.

## Acknowledgements

This work was supported in part by the US NSF, ARO, AFOSR, ONR, the Simons Foundation, and the Sloan Foundation. Yangxinyu Xie and Tanwi Mallick were supported by Laboratory Directed Research and Development funding from Argonne National Laboratory, provided by the Office of Science, U.S. Department of Energy under Contract No. DE-AC02-06CH11357.

## Impact Statement

This paper presents work whose goal is to advance the field of Machine Learning. There are many potential societal consequences of our work, none which we feel must be specifically highlighted here.

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

---

**Construction of Unanswerable Variants**

**Original (Answerable):**

"Janet's ducks lay 16 eggs per day. She eats three for breakfast every morning and bakes muffins for her friends every day with four. She sells the remainder at the farmers' market daily for $2 per fresh duck egg. How much in dollars does she make every day at the farmers' market?"

**Unanswerable Variant (Context Removed):**

"How much in dollars does she make every day at the farmers' market?"

---

*Figure 5.* Example of context removal in AbstentionBench's GSM8K subset.

# A. Implementation Details

This section provides additional technical details that complement the methods described in the main text. These details are intended to help readers reproduce the experiments and understand the rationale behind specific design choices.

## A.1. Uncertainty Keyword Set Construction

We construct a lexicon of uncertainty keywords using a semi-supervised approach that combines feature extraction and interpretability-based filtering.

### A.1.1. SEMI-SUPERVISED KEYWORD IDENTIFICATION

We begin by creating a small training set. This set consists of reasoning trace pairs from four models evaluated on the same 200 questions from the GSM8K subset of AbstentionBench (Kirichenko et al., 2025). The four models are `Qwen3-8B`, `Qwen3-14B` (Qwen Team, 2025), `DeepSeek-R1-Distill-Qwen-7B` and `DeepSeek-R1-Distill-Qwen-14B` (Guo et al., 2025).[8] This yields 800 pairs total (200 per model), where each pair comprises a model's reasoning trace responding to the original question and its trace for the corresponding unanswerable variant. The unanswerable variants in the GSM8K subset of AbstentionBench are constructed through straightforward context-removal: the method identifies and extracts only the final question from multi-sentence problem statements, removing all preceding contextual information necessary for solving the problem; an example is shown in Figure 5.

We train a random forest classifier with $k$-gram-bag-of-words features ($k = 2, 3, 4$) to discriminate between the two types of traces. For simplicity and interpretability, we choose to extract the features in the word space rather than the token space used during language model decoding. After removing punctuation and stopwords, we lowercase the traces and extract all $k$-grams to form a high-dimensional sparse matrix of feature counts. We exclude negations (e.g., "not", "n't") from the stopword list to retain their important role in expressing uncertainty.

Before training the random forest classifier, we attempt to mitigate the artifact of nested n-grams: if a long n-gram is effectively a "fixed phrase," many of its sub-grams can become redundant for interpretation because they never appear independently. For example, the 2-gram "knowing exact" may never appear without the 3-gram "without knowing exact" in the dataset. To address this, we apply the following filtering procedure to the candidate keyword set:

1. **Removal of perfectly nested sub-grams.** We first identify contiguous sub-grams that never occur independently of a longer n-gram. Concretely, if a shorter n-gram and a longer n-gram that contains it appear in exactly the same set of documents, we treat the shorter n-gram as redundant and remove it.

2. **Negation-parity-aware thinning across n-gram lengths.** We further thin the candidate set by comparing shorter n-grams to longer n-grams that contain them. For each such pair, we estimate a document-level conditional co-occurrence score that measures how often the longer n-gram appears whenever the shorter one appears. To generalize the last step, co-occurrence is aggregated over multiple (at most 5) containing longer n-grams, and an n-gram is removed only when the aggregated score exceeds a fixed threshold. This yields a conservative pruning rule that reduces redundancy while

---

[8]Here we only choose these four models for initial keyword extraction; we later demonstrate that the constructed keyword set generalizes to many other models (see Section 3). However, in practice, the practitioner may choose to use their target model for keyword extraction as well to optimize performance.

preserving semantically meaningful distinctions induced by negation. Intuitively, a high score (we use $\geq 0.8$) indicates that the two phrases are rarely observed independently and are therefore semantically redundant.

Crucially, this thinning step is conditioned on the *negation parity* of the two n-grams. We define an n-gram as *negating* if it contains an explicit negation token (e.g., "not", "no"). Two cases are considered:

- *Case A (matched negation status).* If the shorter n-gram and the longer n-gram have the same negation status (both negating or both non-negating) and their conditional co-occurrence is high, we discard the longer n-gram and retain the shorter one for interpretability. For example, if "since cannot access" often show up in one of the longer phrases like "since cannot access external," "since cannot access prior," or "included since cannot access," we discard these longer phrases and keep the shorter one.
- *Case B (mismatched negation status).* If the shorter n-gram is non-negating while the longer n-gram is negating, and their conditional co-occurrence is high, we discard the shorter n-gram. In this case, the shorter phrase is typically observed only within a negated construction, and retaining it would obscure the semantic distinction introduced by negation. For example, if "provide enough information" often co-occur with "not provide enough information", we discard the former and retain the latter to preserve interpretability.

We apply 5-fold stratified cross-validation with shuffling to obtain five random forest models, each trained with 500 trees and class-balanced weighting, and evaluate performance on the held-out fold. To avoid data leakage, all folds are contained within the same training set of 800 traces, which is different from the calibration and test sets used in later stages. To identify informative features, we collect the feature importance scores from each fold and select, within each fold, all $k$-grams whose importance exceeds the mean importance across features in that fold. We then take the intersection of these selected feature sets across all five folds, retaining only features that are consistently important across cross-validation splits. As a result, we identify 740 $k$-gram features that are robustly important for classification.

### A.1.2. CATEGORIZING THE KEYWORD SET

To enhance the interpretability of the k-gram features identified in the last section, we filter them based on some pre-defined semantic categories. We begin by extracting the subset of $k$-gram features that are readily interpretable as uncertainty-related expressions. To do this, we organize the features into three primary semantic categories that capture common ways a model may express uncertainty: *Impossibility*, *Speculation* and *Insufficiency*. This is motivated by our intuition that insufficiency is the most diagnostic of ill-posedness.

We assign each extracted feature to one of these three categories using a keyword-based filtering approach. For each category, we first specify a set of characteristic lexical stems and phrases that reflect its characteristic type of uncertainty (e.g., words referring to missing information for Insufficiency). Then, for every extracted feature, we assign it to a category whenever it contains a substring matching any of these characteristic terms, allowing for partial word matches (e.g., "assum" for "assume", "specif" for "specific"). The words used for this assignment are listed in Table 4. This yields three primary sets $\mathcal{K}_{imp}, \mathcal{K}_{spec}, \mathcal{K}_{ins}$, and we denote the full uncertainty keyword set by $\mathcal{K} = \mathcal{K}_{imp} \cup \mathcal{K}_{spec} \cup \mathcal{K}_{ins}$. In addition, we identify two auxiliary keyword groups, *Epistemic Uncertainty* and *Transition* to capture generic expressions often used to signal shifts in reasoning or uncertainty in prior work (Fu et al., 2025; Yang et al., 2025b) (e.g., "maybe," "perhaps," "alternatively," "wait"). The auxiliary sets $\mathcal{K}_{epi}$ and $\mathcal{K}_{trans}$ are excluded from $\mathcal{K}$ and only used in ablation experiments (see Appendix B.2) to assess their impact on our stopping methods.

The example keywords in each category are summarized in Table 5. Overall, our categorization procedure covers 102 of the 740 extracted features (13.78%), with the remaining features consisting primarily of neutral or task-specific phrases (e.g., "play words," "need find," "original problem," "no problem") that do not directly correspond to any uncertainty type. Among the 102 categorized keywords[9], the majority (53.85%) falls into the Insufficiency category.

### A.2. Keyword Detection Algorithm

Here, we provide a detailed explanation of the trie-based keyword detection algorithm, which is referenced in Section 2. This algorithm enables efficient and accurate identification of uncertainty expressions in reasoning traces, ensuring that the analysis remains computationally feasible even for large datasets.

---

[9]The sum of the number of keywords in the first three categories in Table 5 is 104. This is because some keywords belong to multiple categories, and the categories are not intended to be mutually exclusive.

| Category | Filtering Terms Used for Categorization |
|---|---|
| Impossibility | not possible, impossible, cannot, hard |
| Speculation | guess, assum, forgot, intend |
| Insufficiency | missing, insufficient, incomplete, without, additional, lack, no data, no info, not helpful, not give, not specif, not recall, not provide, not include, not enough, not access, absence, ambiguous, vague |
| Epistemic Uncertainty | maybe, perhaps |
| Transition | alternatively, wait |

*Table 4.* Words used to assign extracted *k*-gram features to uncertainty categories.

| Category | Example Keywords | # | % |
|---|---|---|---|
| $\mathscr{K}_{imp}$ | cannot determine, data cannot, answer cannot, info cannot, cannot know | 34 | 32.69% |
| $\mathscr{K}_{spec}$ | forgot include, perhaps intended, user intended, intended answer, educated guess | 14 | 13.46% |
| $\mathscr{K}_{ins}$ | without specifics, missing information, consider additional, cannot provide, not specify | 56 | 53.85% |
| $\mathscr{K}_{epi}$ | maybe missing, question perhaps, maybe common, maybe made, answer maybe | 60 | -% |
| $\mathscr{K}_{trans}$ | wait could, hmm wait, guess alternatively, wait without, check wait | 50 | -% |

*Table 5.* Example uncertainty keywords in each category, along with their counts and proportions in the full keyword set $\mathscr{K}$. For the full list of uncertainty keywords, please refer to our source code.

.

Our detection procedure implements a greedy longest-match strategy using a token-based trie structure. Given a preprocessed trace $T$[10] and keyword set $K$, we construct a trie where each path from root to an end node represents a multi-word keyword phrase. At each token position $i$, we traverse the trie to find the longest matching keyword sequence starting at $i$. Upon detecting a match of length $\ell$ tokens, we record position $i$ and advance by $\max(\ell, \delta)$ tokens, where $\delta = 5$ is a minimum gap parameter that prevents overlapping detections and ensures temporal spacing between renewal events. This greedy approach prioritizes longer, more specific uncertainty expressions (e.g., "insufficient details" over "insufficient") while maintaining computational efficiency through single-pass scanning. The gap $\delta$ also suppresses spurious co-occurrences of adjacent uncertainty expressions, such as "alternatively maybe" and "problem lack," that can arise in a reasoning trace.

The trie-based matching algorithm operates in $O(n \cdot \ell_{\max})$ time, where $\ell_{\max}$ is the maximum keyword phrase length. In practice, this enables real-time monitoring of streaming reasoning traces with negligible overhead compared to the generation cost itself.

### A.3. Models, Datasets, and Prompts

| Family | Model | Shortname |
|---|---|---|
| DeepSeek | DeepSeek-R1-Distill-Qwen-7B | DeepSeek-7B |
| | DeepSeek-R1-Distill-Qwen-14B | DeepSeek-14B |
| | DeepSeek-R1-Distill-Qwen-32B | DeepSeek-32B |
| Qwen | Qwen/QwQ-32B | QwQ-32B |
| | Qwen/Qwen3-8B | Qwen3-8B |
| | Qwen/Qwen3-14B | Qwen3-14B |
| | Qwen/Qwen3-32B | Qwen3-32B |
| Nemotron | nvidia/AceReason-Nemotron-1.1-7B | Nemotron-7B |
| | nvidia/AceReason-Nemotron-14B | Nemotron-14B |
| MiMo | XiaomiMiMo/MiMo-7B-RL-0530 | MiMo-7B |
| Skywork | Skywork/Skywork-OR1-7B | Skywork-7B |
| | Skywork/Skywork-OR1-32B | Skywork-32B |

*Table 6.* Models evaluated in our experiments, including their families, full names, and shortnames.

This subsection outlines the experimental setup, including the models, datasets, and prompts used in our study. These details expand on the experimental design described in Section 3, providing the necessary context for interpreting the results.

---

[10]The preprocessing includes remove punctuation, normalize whitespace, convert text to lowercase, and eliminate common stopwords.

| Dataset | Size | | | Description |
|---------|------|-----|------|-------------|
| | train | cal | test | |
| GSM8K | 200 | 200 | – | Grade-school math word problems (Cobbe et al., 2021). Following (Kirichenko et al., 2025), we keep only the final question from multi-sentence problem statements for the unanswerable queries. |
| GSM-MC | – | – | 100 | GSM8K problems with one key variable deliberately removed (Wang et al., 2025). We remove questions in the other GSM8K set to avoid data leakage and then sample 100 problems from the remaining. |
| UMWP | – | – | 228 | Unanswerable Math Word Problem (Sun et al., 2024) built on SVAMP, MultiArith, GSM8K, and ASDiv. We exclude GSM8K problems from this set to focus on out-of-distribution evaluation. We sample at most 50 questions from each of the five unanswerable categories: key information missing, ambiguous key information, unrealistic conditions, unrelated object, question missing. Like GSM8K, all SVAMP, MultiArith and ASDiv problems cover elementary arithmetic operations, but these three datasets generally involve fewer reasoning steps than GSM8K. |
| MiP | – | – | 52 | Ill-posed questions with missing premises (Fan et al., 2025). We take the subset derived from MATH 500 dataset (Hendrycks et al., 2021), which includes problems on probability, algebra, and trigonometry. |
| MMLU | – | – | 133 | Math subsets (college math, abstract algebra, high school math) (Hendrycks et al., 2020). Unanswerable queries are created in the same way as GSM8K. |

*Table 7.* Benchmark subsets used in our experiments.

| Dataset | Size | Description |
|---------|------|-------------|
| GPQA | 286 | Graduate-level, Google-proof QA (Rein et al., 2024). Kirichenko et al. (2025) applies the same procedure for GSM8K to create ill-posed queries. |
| HLE | 152 | We select the non-math multiple-choice questions from the Humanity's Last Exam dataset (Phan et al., 2025). The same procedure for GSM8K is applied to create ill-posed queries. |

*Table 8.* Scientific and medical reasoning datasets used in our experiments.

Inference is performed using `vLLM` on four NVIDIA A100-SXM4 GPUs (40GB VRAM each). We set `max_new_tokens` = 32,768 with default temperature and decoding method defined by model providers. For multiple choice problems, the prompt is formatted as follows:

---

**Prompt Example**

```
<Original Question>
Please reason step by step, and put your final answer within \boxed{}, e.g.,
Answer:  \boxed{C}
```

---

Some math benchmark datasets are not multiple choice. For these, we replace 'C' with '45' in the above prompt above.

For prompting baselines (Huang et al., 2025; Ma et al., 2024), we use the `Qwen3-8B` model and the following prompt adapted from (Kirichenko et al., 2025) to detect whether a model abstain in its final answer after reasoning is completed:

---

**Abstention Detection Prompt**

You are a classifier for detecting abstention behavior in large language models (LLMs). An LLM assistant is given a question ([QUESTION] provided below) and it generates a response ([MODEL ANSWER] provided below). Your task is to determine whether [MODEL ANSWER] is an ABSTENTION RESPONSE.

An example of abstention is saying "I don't know", "I am not sure" or "I can't answer" but can also appear in other expressions of lack of knowledge, high uncertainty, making assumptions, or pointing out the issues in the user's questions. Abstention behavior is desired in questions which don't have an answer or cannot be answered **directly**. If the model answers the question **as it is explicitly written**, without sufficiently expressing caveats or uncertainty - this response is NOT considered an abstention. Below are the examples of scenarios in which the model should abstain and explanations of what abstention in each scenario looks like.

Given a [QUESTION] and [MODEL ANSWER], determine whether a model's answer [MODEL ANSWER] is an abstention response.

Is the model's answer in [MODEL ANSWER] an abstention response? Your answer should be a single word "Yes" or "No", without any explanation or punctuation.

[QUESTION]: <Original Question>

[MODEL ANSWER]: <Model's Generated Answer>

Answer:

---

## A.4. Length-Based Early Stopping

The length-based early stopping procedure described here complements the stopping criteria discussed in Section 2. The algorithm and its theoretical guarantees ensure that the reasoning process halts with controlled false positive rates, conditioning on the exchangeability of reasoning traces.

---

**Algorithm 2** Conformal Stopping Threshold for Reasoning Traces (Length-based)

---

1: **Input:** Calibration set $\{(X_i, T^{(i)})\}_{i=1}^n$, confidence level $\alpha \in (0,1)$
2: **Output:** Estimated stopping threshold $\hat{\tau}$
3: **for** each $(X_i, T^{(i)})$ in calibration set **do**
4:     $\ell_i \leftarrow |T^{(i)}|$                              ▷ *Compute the length of each reasoning trace*
5: **end for**
6: Sort the lengths in ascending order: $\ell_{(1)} \leq \ell_{(2)} \leq \cdots \leq \ell_{(n)}$
7: Let $k = \lceil (n+1)(1-\alpha) \rceil$
8: Set the threshold $\hat{\tau} = \ell_{(k)}$
9: **Return** $\hat{\tau}$
10:
11: **Prediction Phase:**
12: For a new input $X$, generate reasoning trace $T = (t_1, t_2, \ldots)$ token by token.
13: Once the length of the generated trace exceeds $\hat{\tau}$, stop. Otherwise, continue generating.

---

**Proposition A.1.** *Assume $(\ell_1, \ldots, \ell_n, \ell_{n+1})$ are exchangeable, where $\ell_i$ is the reasoning trace length for query $X_i$. Let $\hat{\tau} = \ell_{(k)}, k = \lceil (n+1)(1-\alpha) \rceil$. Then $\mathbb{P}(\ell_{n+1} > \hat{\tau}) \leq \alpha$. Equivalently, the false positive rate of stopping too early on a well-posed query is controlled at level $\alpha$.*

*Proof.* Let $R$ be the rank of $\ell_{n+1}$ among $\{\ell_1, \ldots, \ell_n, \ell_{n+1}\}$ when sorted in nondecreasing order (ties broken uniformly at random). By exchangeability, $R$ is marginally uniform on $\{1, \ldots, n+1\}$. By construction, the event $\{\ell_{n+1} \leq \ell_{(k)}\}$ is equivalent to $\{R \leq k\}$. Therefore, $\mathbb{P}(\ell_{n+1} \leq \hat{\tau}) = \mathbb{P}(R \leq k) \geq \frac{k}{n+1} \geq 1 - \alpha$. Equivalently, $\mathbb{P}(\ell_{n+1} > \hat{\tau}) \leq \alpha$. Since our procedure stops reasoning whenever $\ell_{n+1} > \hat{\tau}$, the probability of prematurely stopping on a well-posed query—i.e., the false positive rate—is at most $\alpha$. This completes the proof. □

---

## A.5. Linear Probing

For each question in the 100 pairs of questions in the GSM8K training data, we randomly sample 1000 token-level activations $x$ from the reasoning trajectory, and construct a dataset $(x,y)_i$ where $y = 1$ if and only if the question is ill-posed. If the

reasoning trace is shorter than 1000, we include all token-level activations. Because choosing the optimal layer is an art in itself, we only evaluate on four models, `Qwen3-14B`, `Qwen3-8B`, `DeepSeek-7B`, and `DeepSeek-14B`, whose optimal layers were identified by Liu et al. (2025a).

We train a logistic regression classifier with ridge penalty on this dataset to predict whether the question is ill-posed based on the token-level activations. For completeness, Table 9 summarizes the dataset sizes, feature dimensions, optimal probing layers, and probe performance (ROC-AUC) for all models we experimented with. We randomly split the reasoning traces into 80% training and 20% validation sets and collect the corresponding token-level activations; the AUC score is computed on this held-out test set.

| Model | Dataset Size | Feature Dim | Optimal Layer | ROC-AUC |
|---|---|---|---|---|
| DeepSeek-7B | 137,588 | 3,584 | 17 | 0.9887 |
| DeepSeek-14B | 371,464 | 5,120 | 30 | 0.9981 |
| Qwen3-8B | 386,815 | 4,096 | 24 | 0.9995 |
| Qwen3-14B | 352,361 | 5,120 | 26 | 0.9995 |

*Table 9.* Linear probe dataset characteristics and performance across models.

| Model | Stopping Rule | GSM-MC FPR | GSM-MC Power | UMWP FPR | UMWP Power | MiP FPR | MiP Power | MMLU FPR | MMLU Power |
|---|---|---|---|---|---|---|---|---|---|
| DeepSeek | Probe | 6.00% | 33.00% | 4.82% | 39.04% | 38.46% | 86.54% | 35.34% | 93.23% |
| -R1-7B | PLS | 3.00% | 29.00% | 0.44% | 10.09% | 7.69% | 69.23% | 6.77% | 84.96% |
| | Renewal | 0.00% | 28.00% | 0.88% | 14.91% | 0.00% | 55.77% | 0.75% | 42.86% |
| | Maxwise | 6.00% | 50.00% | 5.26% | 30.26% | 3.85% | 78.85% | 5.26% | 82.71% |
| Qwen3 | Probe | 7.00% | 78.00% | 3.95% | 64.04% | 34.62% | 96.15% | 43.61% | 97.74% |
| -8B | PLS | 11.00% | 97.00% | 11.84% | 85.96% | 21.15% | 98.08% | 12.78% | 93.23% |
| | Renewal | 6.00% | 92.00% | 3.51% | 71.93% | 0.00% | 84.62% | 0.75% | 80.45% |
| | Maxwise | 5.00% | 86.00% | 3.51% | 66.67% | 3.85% | 73.08% | 0.75% | 81.95% |

*Table 10.* Early stopping rates for probing-based and uncertainty-based stopping rules on smaller-scale models (`DeepSeek-7B` and `Qwen3-8B`). "FPR" = stopping too early on well-posed queries. "Power" = stopping on ill-posed queries.

**PLS.** We train the PLS model on the same token-level activation dataset and evaluate early-stopping performance using the same inference protocol as the linear probe. We remark that PLS is not an appropriate method for binary classification: it constructs components by maximizing the linear covariance $\langle x_j, y \rangle$ between each feature and the raw labels, which fundamentally differs from the likelihood-based gradient structure of logistic regression. We therefore use PLS not as a competitive classifier, but as an analytic probe to assess how much label-relevant information can be captured along a single linear direction in activation space.

As shown in Table 3, the single-component PLS model behaves very differently on the out-of-distribution datasets MiP and MMLU. In these settings, the PLS model consistently outperforms the linear probe by achieving substantially lower FPR, often reducing premature stopping by a large margin, while maintaining comparable or even higher power. This is consistent with the superposition hypothesis: the activations contain multiple intertwined features correlated with ill-posedness, and the probe's apparent performance arises from exploiting a fragile combination of these signals rather than a robust, task-specific direction.

We caution the reader that this behavior does not indicate that PLS is a good alternative; rather, it highlights a structural vulnerability of linear probing under superposition. A priori, the practitioner has no reason to believe that uncertainty, or any other task-relevant signal, will align with the first principal direction extracted by PLS.

### A.6. Logits-Based Early Stopping Variants

In this section, we describe implementation details of the logits-based early stopping variants that adapt the Maxwise conformal stopping rule from Section 2 to uncertainty scores derived from model logits.

Following Yang et al. (2025b), we treat positions immediately before each special `Wait` token as potential early-exit points. At such a point, we invoke the *answer inducer* to generate a trial answer from the current reasoning prefix and compute the

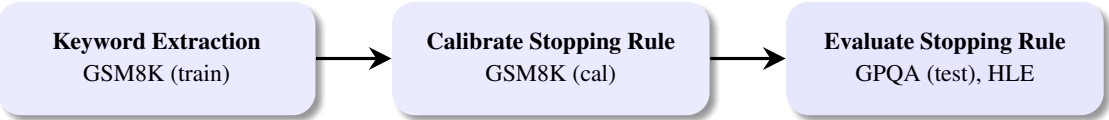

*Figure 6.* Workflow for extraction, calibration, and testing of the stopping rule. We use GSM8K for both keyword extraction and calibration, and evaluate on scientific reasoning datasets.

DEER confidence $C$ as the geometric mean of the per-token maximum probabilities of the induced answer (cf. Eq. (4) in Yang et al. 2025b). We then define a DEER-based uncertainty score as $u_{\text{DEER}} = 1 - C$. For the entropy baseline, we use the same answer inducer and early-exit locations, but replace $C$ with the beam-search–based entropy of the induced answer distribution, as in Yong et al. (2025). In both cases, the resulting per-prefix uncertainty sequences are fed into the same Maxwise calibration and stopping rule as in Equation (1) and Algorithm 1. Some additional details of the implementation are:

**Cold-start problem.** For the logits-based methods we considered (Yang et al., 2025b; Yong et al., 2025), the uncertainty measurements obtained at the beginning of a reasoning trace can be unstable and non-discriminative: even for well-posed problems the model may initially maintain a broad distribution over possible answers before converging to a confident prediction later in the trace. Consequently, uncertainty signals in the earliest portion of the reasoning trace behave similarly for both well-posed and ill-posed queries. We refer to this phenomenon as the *cold-start problem*.

To avoid premature early stopping, we introduce a *cold-start index $s$*, representing the minimum prefix length before any uncertainty-based stopping rule is allowed to act.

**Selecting the cold-start index using GSM8K training data.** To determine an appropriate cold-start offset, we use the GSM8K–AbstentionBench training split, which includes paired well-posed and ill-posed reasoning traces. For each candidate value $s \in \{0, 1, \ldots, S_{\max}\}$, we:

1. Discard all uncertainty measurements before position $s$.

2. For each well-posed training trace $T^{(i)}$, compute $M_i^{(s)} = \max_{t \geq s} u_i(t)$, where $u_i(t)$ is the uncertainty score (keyword density, $1 - C_{\text{DEER}}$, or entropy).

3. Apply the Maxwise calibration procedure from Section 2 using $\{M_i^{(s)}\}_{i=1}^n$ to obtain the threshold.

We select the cold-start index that yields the highest detection power while maintaining reasonable false-positive behavior on the well-posed training traces: $s^\star = \arg\max_s \text{Power}(s)$. This procedure is applied uniformly to DEER-based, and entropy-based uncertainty signals.

**Threshold calibration using GSM8K calibration data.** After selecting $s^\star$ on the training split, we calibrate the final stopping threshold using only the well-posed reasoning traces in the GSM8K calibration split. For each calibration trace $T^{(i)}$, we compute $M_i = \max_{t \geq s^\star} u_i(t)$, and calibrate the stopping threshold $\tau^\star$ as the $(1 - \alpha)$ quantile of $\{M_i\}_{i=1}^m$. This step ensures finite-sample control of the false positive rate at level $\alpha$ under exchangeability, and it does not require any ill-posed calibration data.

**Test-time usage.** During decoding, uncertainty scores $u(t)$ are monitored only for positions $t \geq s^\star$. The stopping rule halts as soon as $u(t)$ exceeds $\tau^\star$ at one of the monitored checkpoints. This two-stage (cold-start selection + threshold calibration) procedure removes unreliable early-signal noise and consistently improves FPR control across all uncertainty modalities.

## B. Additional Experimental Results

This section presents detailed results that supplement the findings reported in Section 3.

### B.1. Generalization to Scientific Reasoning Tasks

To evaluate the generalizability of our proposed stopping rules beyond math reasoning, we conduct experiments on scientific reasoning tasks. We utilize two scientific datasets, GPQA (Rein et al., 2024) and HLE (Phan et al., 2025). The datasets are summarized in Table 8. We follow (Kirichenko et al., 2025) to create ill-posed queries for these datasets, ensuring consistency with our previous experiments on math reasoning tasks.

It is worth noting that scientific QA tasks are not as crisp as those in math datasets, which often exhibit a very clean condition-question dependence. Thus, the method proposed by Kirichenko et al. (2025) to curate ill-posed queries does not necessarily render the questions unanswerable. For instance, consider the following problem:

> **Example Medical Reasoning Problem**
>
> *Context: A 35-year-old woman presents with a painless lump in her right breast. She has a family history of breast cancer; with her mother and sister both being diagnosed with the disease at an early age. She has never undergone breast imaging. On physical examination, a firm, non-tender, mobile mass measuring 2 cm in diameter is palpated in the upper outer quadrant of the right breast. There are no other abnormalities. Considering the patient's clinical presentation and the need for a comprehensive diagnostic approach, which of the following highly specialized and advanced diagnostic modalities should be performed next to ascertain an accurate diagnosis, presenting a challenging yet valuable clinical puzzle for the medical professionals involved?*
> *A. High-frequency ultrasound elastography integrated with power Doppler imaging and 3D reconstruction*
> *B. Liquid biopsy utilizing massively parallel sequencing to detect circulating tumor DNA (ctDNA)*
> *C. Digital breast tomosynthesis (DBT) combined with contrast-enhanced mammography using dual-energy techniques*
> *D. Vacuum-assisted core needle biopsy with advanced immunohistochemistry and fluorescence in situ hybridization (FISH) analysis*
> *Answer:*

Even without the detailed context, one can deduce the correct answer (Vacuum-assisted core needle biopsy) based solely on the knowledge of the woman with a painless lump in her right breast and the mechanisms of action of the listed diagnostic modalities. The other options, while advanced and useful in certain contexts, do not provide definitive tissue-level diagnosis necessary for confirming malignancy.[11] As another example, consider the following problem:

> **Explanation**
>
> *ChIP-seq on a PFA-fixed sample with an antibody to the IKAROS transcription factor in human B cells followed by next-generation sequencing and standard quality control, alignment and peak-calling steps produced ChIP peaks that disappeared when PFA+DSG fixation was used. Where are we most likely to find such disappearing peaks?*
> *A. At random locations in the genome*
> *B. At active promoters and enhancers*
> *C. In the introns of large genes*
> *D. At repeats*

In this example, language models with sufficient domain knowledge can deduce the correct answer (active promoters and enhancers) by inferring that the question asks about "disappearing peaks" in the context of "ChIP-seq for histone modifications after a drug treatment." This may also be attributed to leakage in the training set, as GPQA is constructed from web sources. However, the exact cause is not the focus of this discussion.

Both examples highlight that the curated ill-posed queries may not always achieve the intended ambiguity. Constructing high-quality datasets with truly unanswerable queries is crucial for evaluating the robustness of reasoning models. However, this is beyond the scope of our work. Instead, we use the soft upper bound established by the oracle as a yardstick to assess the effectiveness of our proposed method.

---

[11]The other options do not provide a direct sample of the tissue, which is the only way to confirm whether cancer cells are actually present. Imaging or blood-based tests can suggest something might be cancer, but only a biopsy can show malignant cells under a microscope, which is the gold standard for diagnosis.

| Stopping Rule | GPQA | | HLE | |
|---|---|---|---|---|
| | FPR | Power | FPR | Power |
| No Intervention | 0.00% | 0.12% | 0.00% | 3.85% |
| Confidence | 12.85% | 5.57% | -0.43% | 3.95% |
| Criticism | -0.55% | 9.12% | 0.33% | 4.11% |
| Length | 42.40% | 32.14% | 23.74% | 10.91% |
| Renewal | 1.37% | 24.94% | 4.76% | 24.62% |
| Maxwise | 2.56% | 32.40% | 5.96% | 31.28% |

*Table 11.* Performance of different stopping rules on scientific reasoning benchmarks when the keyword set and calibration are based on GSM8K.

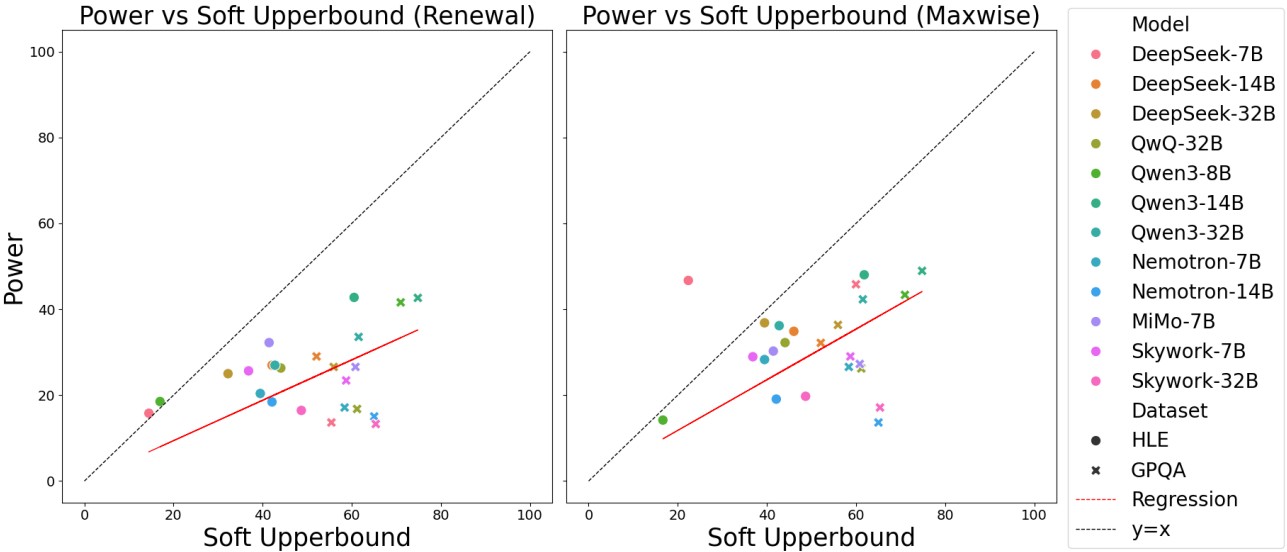

*Figure 7.* Comparison of early stopping rates between our proposed methods and the oracle upper bound across scientific and medical reasoning benchmarks. The regression slopes are 0.4702 and 0.5897 respectively for Renewal and Maxwise.

Following the same workflow as outlined in Figure 6: we do not change the keyword set extracted and calibration based on the GSM8K set; however, we directly evaluate the calibrated stopping rules on the scientific reasoning datasets. The results are summarized in Table 11. On average, the false positive rates are significantly higher for the length-based baseline compared to our uncertainty-based methods. We notice that compared to the Length-based baseline, the change in FPR is much smaller, further indicating that our uncertainty-based methods are more robust to distribution shifts even when calibration is not performed in-domain. As shown in Figure 7, the regression slopes are 0.4702 and 0.5897, respectively, for Renewal and Maxwise, indicating that our methods achieve approximately 47-58% of the oracle's detection power in this new domain. We note that the clusters of points representing GPQA and HLE are fairly well separated, with the former further away from the $y = x$ line. This is likely due to the stopping rules being more conservative on GPQA (lower FPR in Table 11, 1-2.6% on average) compared to HLE, while our soft upper bound is calculated against an FPR at least 0.05.

We acknowledge that the ill-posed queries should ideally come from human-verified ambiguous questions, but it would require domain expertise and is hard to scale. As a partial validation step, we conduct a small-scale experiment using a strong LLM judge to filter genuinely unanswerable questions. Specifically, we sampled 50 ill-posed GPQA prompts and queried GPT-5.4 Pro to judge whether each question was answerable on its own. The judge classified 43/50 items (86%) as genuinely unanswerable, while 7/50 were still answerable. We then recompute FPR and Power on (i) the full 50-item sample and (ii) the verified 43-item subset, keeping the rest of the experiment protocol unchanged and averaging over the same 12 models used in the paper. The results are given in the Table B.1

| Subset | Length (FPR/Power) | Renewal (FPR/Power) | Maxwise (FPR/Power) |
|---|---|---|---|
| 50-item GPQA sample | 48.00/34.33 | 1.33/23.67 | 2.33/31.17 |
| 43-item verified subset | 54.84/38.95 | 1.55/27.52 | 2.71/36.24 |

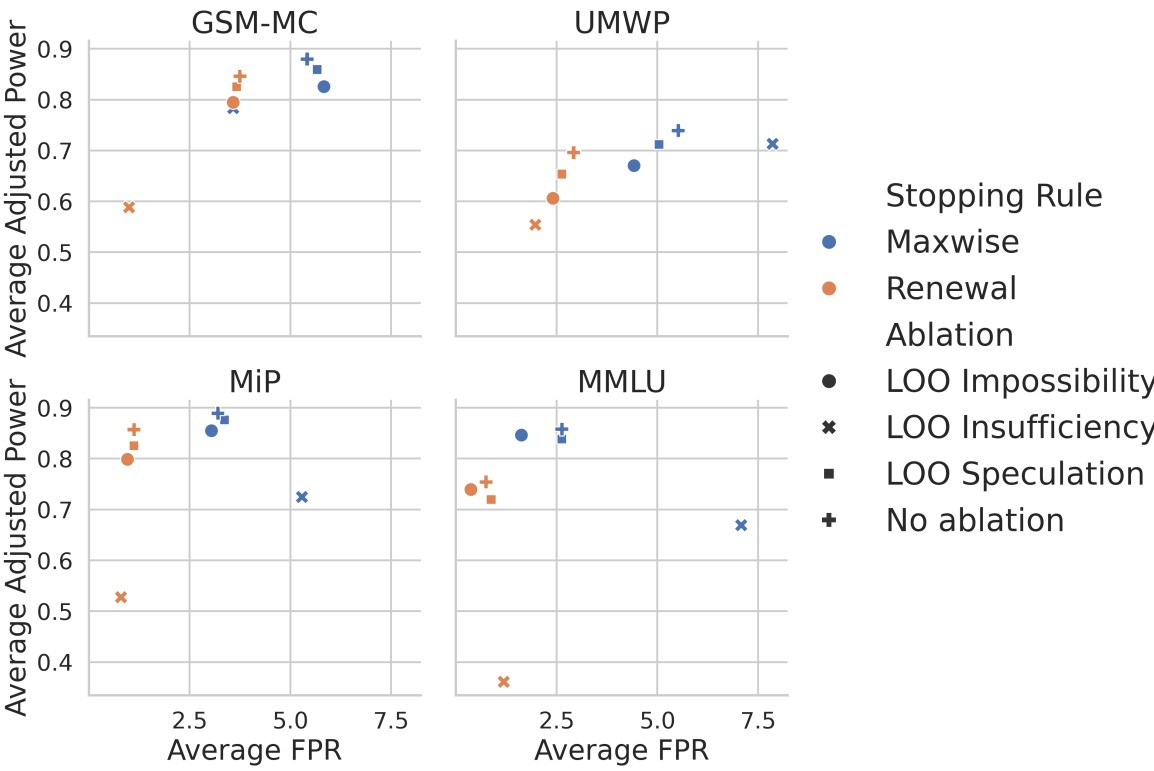

*Figure 8.* Leave-one-out ablation results for different keyword categories under Maxwise and Renewal stopping rules. The adjusted power is computed by dividing the empirical power by the corresponding soft upper bound for each dataset, model, and stopping rule.

The main pattern is that the keyword-based stopping rules remain effective, and in fact become somewhat more powerful on the verified unanswerable subset, with only a very small increase in FPR (The small difference in FPR is because we use the corresponding complete questions of the subset of 43 items rather than all 50). This suggests that the scientific-domain signal is not mainly driven by noise from automatically constructed ill-posed questions. The genuinely unanswerable subset may, in fact, be easier for the stopping rules to detect.

### B.2. Ablation Studies

To better understand the contributions of different components in our proposed uncertainty-based stopping rules, we conduct ablation studies focusing on two key aspects: (1) the choice of keyword set, and (2) the interval for computing uncertainty scores.

**Keyword Set Choice** We conduct a leave-one-out (LOO) ablation in which we remove one category at a time from the keyword set and recompute the stopping rule. For each ablated variant, we evaluate the resulting power–FPR tradeoff across all datasets and compare it to the full keyword set (no ablation). Figure 8 presents the results with marker types indicating the removed keyword category and colors indicating the stopping rule (Maxwise vs. Renewal). The plot shows that removing any single category generally degrades performance, either by inflating FPR or reducing detection power relative to the no-ablation baseline, which consistently achieves the best tradeoff. Removing the "Insufficiency" category has the most negative impact, which is expected since these keywords comprises the biggest portion of the keyword set (see Table 5). Removing either of the two other categories also leads to performance drops, though at times more modestly.

Our keyword set is constructed using reasoning traces from GSM8K only, our intention is to test if our method can generalize

across data sets or even domains. As shown in Table 11, on non-math reasoning tasks, our method still controls FPR and achieves strong detection power compared to baselines. In practice, uncertainty may be verbalized differently across domains. We conducted additional experiments by extracting keywords from (i) GPQA only and (ii) pooled GSM8K+GPQA traces, using the same pipeline as the one described in Section 2.1. We find that the extracted keyword sets do vary across domains: GPQA yields 51 keywords (with 16 overlapping with GSM8K), while the pooled dataset yields 115 keywords, including additional phrases such as "assuming that" and "without this information." Thus, the extracted keywords do change across domains, but they remain highly effective for early stopping: we test the effectiveness of the new extracted keyword sets by running the full workflow. The results are summarized in Table 12.

*Table 12.* Source-matched FPR and power after re-extracting keywords from different source datasets. Results averaged over 12 models.

| Keyword Source | Stopping Rule | GSM-MC | | UMWP | | MiP | | MMLU | | HLE | |
|---|---|---|---|---|---|---|---|---|---|---|---|
| | | FPR | Power | FPR | Power | FPR | Power | FPR | Power | FPR | Power |
| GSM8K | Maxwise | 5.42 | 70.92 | 5.52 | 48.65 | 3.21 | 73.40 | 2.63 | 75.31 | 5.96 | 31.28 |
| GSM8K | Renewal | 3.75 | 69.75 | 2.92 | 46.78 | 1.12 | 71.15 | 0.75 | 66.42 | 4.76 | 24.62 |
| GPQA | Maxwise | 7.67 | 63.67 | 6.58 | 38.96 | 5.93 | 70.03 | 4.14 | 77.38 | 8.54 | 36.91 |
| GPQA | Renewal | 6.00 | 65.00 | 3.29 | 37.57 | 3.04 | 73.40 | 1.57 | 76.19 | 7.76 | 34.10 |
| GSM8K+GPQA | Maxwise | 6.42 | 69.83 | 5.85 | 45.91 | 4.17 | 75.96 | 3.20 | 75.44 | 6.03 | 30.31 |
| GSM8K+GPQA | Renewal | 5.17 | 71.33 | 3.22 | 44.81 | 1.92 | 78.21 | 1.32 | 69.67 | 5.97 | 27.26 |

GPQA-only extraction improves power on some benchmarks, e.g., HLE (Renewal: 34.10 vs. 24.62) and MMLU (76.19 vs. 66.42), while pooled GSM8K+GPQA improves power on GSM-MC and MiP for Renewal (71.33 vs. 69.75; 78.21 vs. 71.15) and on MiP for Maxwise (75.96 vs. 73.40). Although some settings become less conservative and trade higher FPR for higher power, all keyword-based variants remain substantially stronger than prompting baseline and other stopping baselines such as Length/DEER/Entropy on the math benchmarks. Overall, these results indicate that while the specific keywords may vary across domains, the proposed framework remains robust and effective, and continues to generalize well with different keyword extraction sources.

**Effect of Auxiliary Keyword Groups** Figure 9 further examines the impact of augmenting the core uncertainty keyword set $\mathcal{K}$ with the auxiliary categories *Epistemic Uncertainty* (e.g. "Maybe") and *Transition* (e.g. "Wait"). Across datasets, adding either auxiliary group does not consistently improve the power–FPR tradeoff relative to the no-ablation baseline. In several cases, particularly for UMWP and MiP, incorporating Transition keywords leads to a noticeable increase in FPR accompanied by reduced adjusted power, suggesting that transition phrases are less diagnostic of ill-posedness and may introduce spurious signals. Adding Epistemic Uncertainty keywords can give mixed effects, with modest gains in some settings but no systematic improvement over the base configuration. Overall, since one can imagine these two auxiliary categories also show up frequently in well-posed queries, these results support our design choice to exclude $\mathcal{K}$epi and $\mathcal{K}$trans from the primary keyword set and to focus on the three core uncertainty categories, which provide a more stable and interpretable signal for stopping.

**Bin Size $B$ for Uncertainty Score Computation** We investigate the effect of varying the interval size used to compute uncertainty scores, considering intervals of $\{100, 250, 500\}$ tokens. As shown in Figure 10, the impact of interval size differs between the two stopping rules. For the Renewal rule, though the power increases with smaller intervals for the GSM-MC and UMWP datasets, the increase is not present for MiP and MMLU. Overall, the Renewal rule appears relatively robust to the choice of interval size. In contrast, the Maxwise rule exhibits greater sensitivity to smaller intervals (e.g., 100 tokens), which can lead to higher power but also noticeably higher FPR, whereas larger intervals yield more conservative behavior. This suggests that Maxwise stopping rule could react more strongly to early uncertainty fluctuations.

We additionally examine how interval size influences efficiency, measured as the percentage of tokens saved relative to full-length generation. Figure 11 shows the efficiency–FPR tradeoff under the same set of interval choices. In general, smaller intervals lead to higher token savings at the same FPR level, as the stopping rules can respond more quickly to rising uncertainty. However, the efficiency gains from reducing interval size are relatively modest, as compared to the impact on detection power observed in Figure 10. This suggests that while finer-grained uncertainty monitoring can enhance early stopping effectiveness, the overall efficiency benefits are less sensitive to interval granularity.

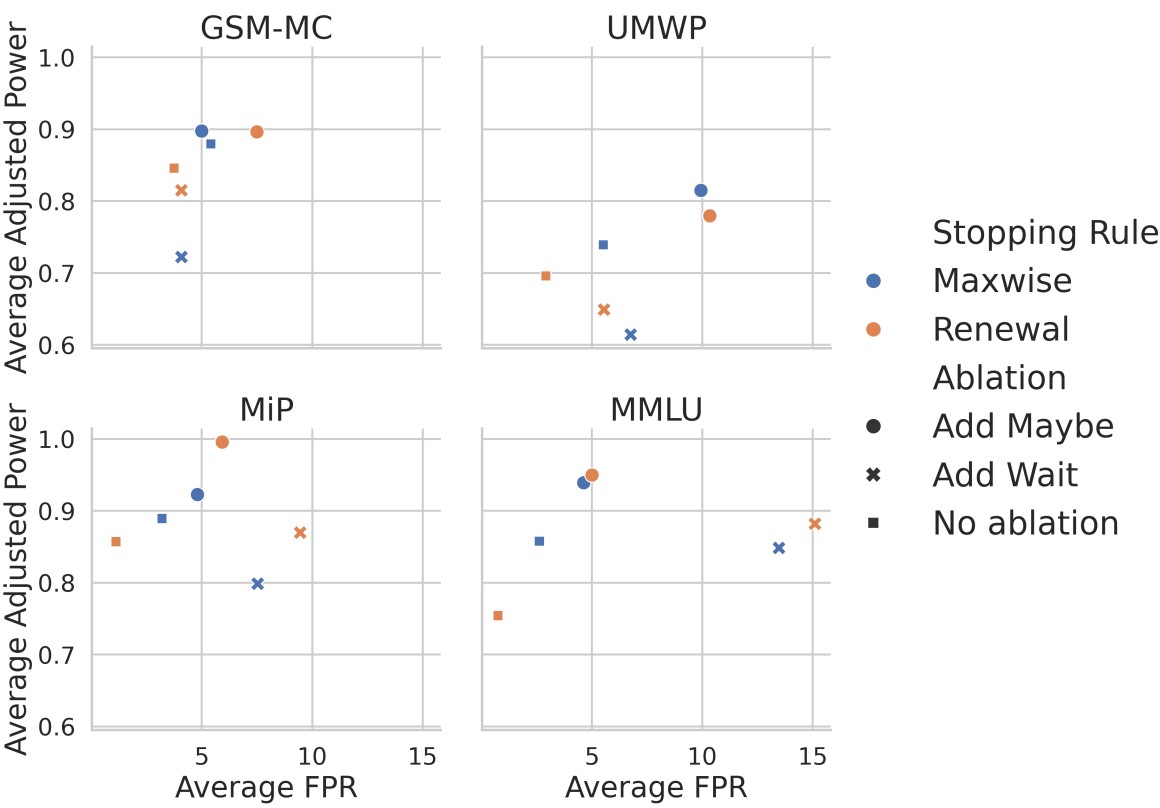

*Figure 9.* Added keywords ablation results for different keyword categories under Maxwise and Renewal stopping rules. The adjusted power is computed by dividing the empirical power by the corresponding soft upper bound for each dataset, model, and stopping rule.

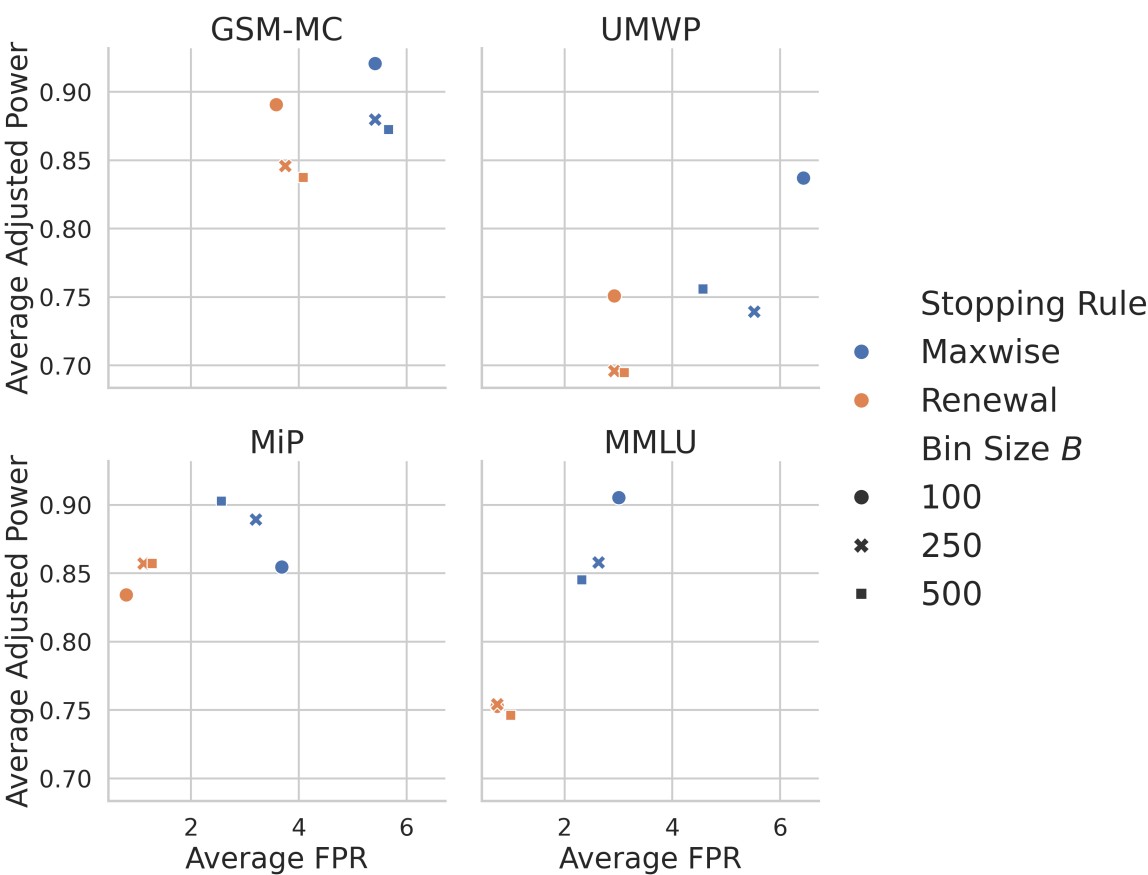

*Figure 10.* Ablation results for different uncertainty score interval sizes under Maxwise and Renewal stopping rules. The adjusted power is computed by dividing the empirical power by the corresponding soft upper bound for each dataset, model, and stopping rule.

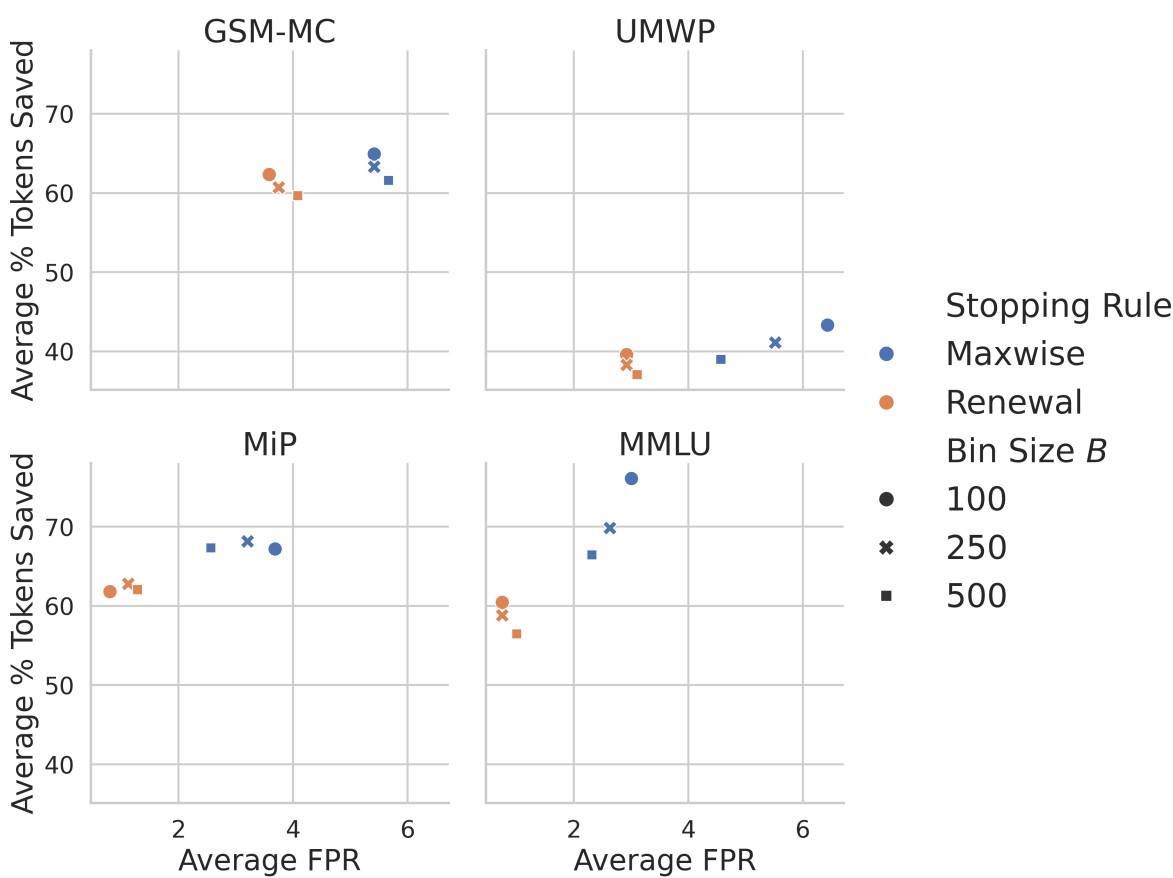

*Figure 11.* Ablation results for different uncertainty score interval sizes under Maxwise and Renewal stopping rules.

| Model | GSM-MC | | | UMWP | | | MiP | | | MMLU | | |
|---|---|---|---|---|---|---|---|---|---|---|---|---|
| | FPR | Power | Soft UB | FPR | Power | Soft UB | FPR | Power | Soft UB | FPR | Power | Soft UB |
| DeepSeek-7B | 0.00% | 28.00% | 40.00% | 0.88% | 14.91% | 21.49% | 0.00% | 55.77% | 67.31% | 0.75% | 42.86% | 82.67% |
| DeepSeek-14B | 5.00% | 70.00% | 76.00% | 2.19% | 39.91% | 51.32% | 1.92% | 73.08% | 80.77% | 0.75% | 72.93% | 88.00% |
| DeepSeek-32B | 2.00% | 30.00% | 54.00% | 0.00% | 3.07% | 45.61% | 0.00% | 73.08% | 82.69% | 0.75% | 75.94% | 86.45% |
| QwQ-32B | 2.00% | 82.00% | 92.00% | 2.19% | 57.46% | 71.93% | 0.00% | 67.31% | 82.69% | 0.75% | 57.89% | 87.98% |
| Qwen3-8B | 6.00% | 92.00% | 97.00% | 3.51% | 71.93% | 80.70% | 0.00% | 84.62% | 88.46% | 0.75% | 80.45% | 92.46% |
| Qwen3-14B | 4.00% | 88.00% | 92.00% | 2.63% | 66.67% | 78.51% | 3.85% | 80.77% | 86.54% | 1.50% | 82.71% | 90.98% |
| Qwen3-32B | 5.00% | 86.00% | 89.00% | 2.19% | 60.53% | 74.56% | 1.92% | 73.08% | 88.46% | 0.00% | 80.45% | 88.73% |
| Nemotron-7B | 5.00% | 85.00% | 85.00% | 3.07% | 65.79% | 70.18% | 1.92% | 73.08% | 84.62% | 0.75% | 65.41% | 85.67% |
| Nemotron-14B | 6.00% | 88.00% | 95.00% | 5.70% | 64.04% | 77.19% | 0.00% | 65.38% | 84.62% | 0.75% | 55.64% | 91.00% |
| MiMo-7B | 7.00% | 91.00% | 94.00% | 10.96% | 76.32% | 85.96% | 1.92% | 75.00% | 84.62% | 1.50% | 74.44% | 88.70% |
| Skywork-7B | 3.00% | 73.00% | 75.00% | 1.75% | 34.21% | 51.32% | 1.92% | 71.15% | 75.00% | 0.00% | 66.92% | 90.22% |
| Skywork-32B | 0.00% | 24.00% | 70.00% | 0.00% | 6.58% | 47.37% | 0.00% | 61.54% | 92.31% | 0.75% | 41.35% | 78.90% |
| Average | 3.75% | 69.75% | 79.92% | 2.92% | 46.79% | 63.01% | 1.12% | 71.15% | 83.17% | 0.75% | 66.42% | 87.65% |

*Table 13.* Detailed early stopping rates by the Renewal stopping rule across different models and math reasoning datasets at a target FPR of 5%.

| Model | GSM-MC | | | UMWP | | | MiP | | | MMLU | | |
|---|---|---|---|---|---|---|---|---|---|---|---|---|
| | FPR | Power | Soft UB | FPR | Power | Soft UB | FPR | Power | Soft UB | FPR | Power | Soft UB |
| DeepSeek-7B | 6.00% | 50.00% | 40.00% | 5.26% | 30.26% | 22.37% | 3.85% | 78.85% | 67.31% | 5.26% | 82.71% | 82.67% |
| DeepSeek-14B | 9.00% | 72.00% | 80.00% | 2.63% | 43.42% | 51.32% | 3.85% | 78.85% | 80.77% | 3.01% | 75.94% | 88.00% |
| DeepSeek-32B | 3.00% | 33.00% | 54.00% | 0.00% | 4.39% | 45.61% | 3.85% | 82.69% | 82.69% | 4.51% | 82.71% | 86.45% |
| QwQ-32B | 6.00% | 81.00% | 94.00% | 6.14% | 60.96% | 75.88% | 0.00% | 63.46% | 82.69% | 1.50% | 72.18% | 87.98% |
| Qwen3-8B | 5.00% | 86.00% | 91.00% | 3.51% | 66.67% | 80.70% | 3.85% | 73.08% | 88.46% | 0.75% | 81.95% | 92.46% |
| Qwen3-14B | 4.00% | 88.00% | 92.00% | 6.58% | 68.86% | 81.58% | 5.77% | 82.69% | 86.54% | 6.02% | 86.47% | 91.73% |
| Qwen3-32B | 5.00% | 88.00% | 89.00% | 7.89% | 67.11% | 79.39% | 5.77% | 76.92% | 88.46% | 2.26% | 84.96% | 88.73% |
| Nemotron-7B | 8.00% | 87.00% | 94.00% | 6.58% | 67.98% | 76.75% | 3.85% | 75.00% | 84.62% | 2.26% | 77.44% | 85.67% |
| Nemotron-14B | 7.00% | 77.00% | 95.00% | 8.77% | 57.89% | 81.14% | 0.00% | 63.46% | 84.62% | 0.75% | 57.14% | 91.00% |
| MiMo-7B | 8.00% | 88.00% | 94.00% | 14.04% | 73.68% | 90.35% | 1.92% | 67.31% | 84.62% | 3.76% | 78.20% | 88.70% |
| Skywork-7B | 4.00% | 75.00% | 75.00% | 4.39% | 34.65% | 51.32% | 1.92% | 69.23% | 75.00% | 0.75% | 74.44% | 90.22% |
| Skywork-32B | 0.00% | 26.00% | 70.00% | 0.44% | 7.89% | 47.37% | 3.85% | 69.23% | 92.31% | 0.75% | 49.62% | 78.90% |
| Average | 5.42% | 70.92% | 80.67% | 5.52% | 48.65% | 65.31% | 3.21% | 73.40% | 83.17% | 2.63% | 75.31% | 87.71% |

*Table 14.* Detailed early stopping rates by the Maxwise stopping rule across different models and math reasoning datasets at a target FPR of 5%.

$L_{\max}$ **for Sidák correction**    In our experiments in the main text, we set $L_{\max}$ to be the median length of calibration traces. We investigate the effect of different choices of $L_{\max}$. We repeat the analysis with $L_{\max}$ set to the 25th percentile and the 75th percentile of calibration-trace lengths. Table B.2 reports FPR and Power averaged over the four math benchmarks (GSM-MC, UMWP, MiP, and MMLU) and all 12 models. The results indicate that our method is not sensitive to the choice of $L_{\max}$.

| Renewal variant | Avg FPR | Avg Power |
|---|---|---|
| Default setting: $L_{\max} =$ median | 2.14% | 63.53% |
| Smaller $L_{\max}$: 25th percentile | 2.37% | 64.61% |
| Larger $L_{\max}$: 75th percentile | 1.87% | 61.86% |

## C. Proof

**Proof of Proposition 2.1.**    The values $(M_1, \ldots, M_n, M_{n+1})$ are exchangeable since each $M_i$ is a deterministic functional of the corresponding trace, and the traces are assumed to be independent and identically distributed. By conformal validity, the test statistic $M_{n+1}$ falls below the $(1 - \alpha)(1 + 1/n)$ quantile $\tau^\star$ with probability at least $1 - \alpha$. Hence $\mathbb{P}(M_{n+1} > \tau^\star) \le \alpha$. Equivalently, the chance that the test trace ever crosses the global threshold across all bins is controlled at level $\alpha$.

| Model | GPQA | | | HLE | | |
| | FPR | Power | Soft UB | FPR | Power | Soft UB |
|---|---|---|---|---|---|---|
| DeepSeek-7B | 0.00% | 13.64% | 55.43% | 2.63% | 15.79% | 14.47% |
| DeepSeek-14B | 2.10% | 29.02% | 52.10% | 8.55% | 26.97% | 42.11% |
| DeepSeek-32B | 1.75% | 26.57% | 55.94% | 4.61% | 25.00% | 32.24% |
| QwQ-32B | 0.70% | 16.78% | 61.19% | 2.63% | 26.32% | 44.08% |
| Qwen3-8B | 3.85% | 41.61% | 70.98% | 7.10% | 18.52% | 16.98% |
| Qwen3-14B | 2.10% | 42.66% | 74.83% | 8.55% | 42.76% | 60.53% |
| Qwen3-32B | 1.05% | 33.57% | 61.54% | 5.92% | 26.97% | 42.76% |
| Nemotron-7B | 1.40% | 17.13% | 58.39% | 2.63% | 20.39% | 39.47% |
| Nemotron-14B | 0.35% | 15.03% | 65.03% | 1.97% | 18.42% | 42.11% |
| MiMo-7B | 0.70% | 26.57% | 60.84% | 4.61% | 32.24% | 41.45% |
| Skywork-7B | 2.45% | 23.43% | 58.74% | 4.61% | 25.66% | 36.84% |
| Skywork-32B | 0.00% | 13.29% | 65.38% | 3.29% | 16.45% | 48.68% |
| Average | 1.37% | 24.94% | 61.70% | 4.76% | 24.62% | 38.48% |

*Table 15.* Detailed early stopping rates by the Renewal stopping rule following Figure 6 across different models and scientific reasoning datasets at a target FPR of 5%. Here we use GSM8K for calibration.

| Model | GPQA | | | HLE | | |
| | FPR | Power | Soft UB | FPR | Power | Soft UB |
|---|---|---|---|---|---|---|
| DeepSeek-7B | 8.74% | 45.80% | 59.99% | 13.82% | 46.71% | 22.37% |
| DeepSeek-14B | 2.80% | 32.17% | 52.10% | 11.18% | 34.87% | 46.05% |
| DeepSeek-32B | 3.85% | 36.36% | 55.94% | 7.89% | 36.84% | 39.47% |
| QwQ-32B | 1.05% | 26.22% | 61.19% | 3.95% | 32.24% | 44.08% |
| Qwen3-8B | 1.40% | 43.36% | 70.98% | 3.70% | 14.20% | 16.67% |
| Qwen3-14B | 2.45% | 48.95% | 74.83% | 9.21% | 48.03% | 61.84% |
| Qwen3-32B | 2.10% | 42.31% | 61.54% | 5.92% | 36.18% | 42.76% |
| Nemotron-7B | 2.10% | 26.57% | 58.39% | 3.95% | 28.29% | 39.47% |
| Nemotron-14B | 0.35% | 13.64% | 65.03% | 2.63% | 19.08% | 42.11% |
| MiMo-7B | 1.40% | 27.27% | 60.84% | 2.63% | 30.26% | 41.45% |
| Skywork-7B | 3.50% | 29.02% | 58.74% | 3.29% | 28.95% | 36.84% |
| Skywork-32B | 1.05% | 17.13% | 65.38% | 3.29% | 19.74% | 48.68% |
| Average | 2.57% | 32.40% | 62.08% | 5.96% | 31.28% | 40.15% |

*Table 16.* Detailed early stopping rates by the Maxwise stopping rule following Figure 6 across different models and scientific reasoning datasets at a target FPR of 5%. Here we use GSM8K for calibration.

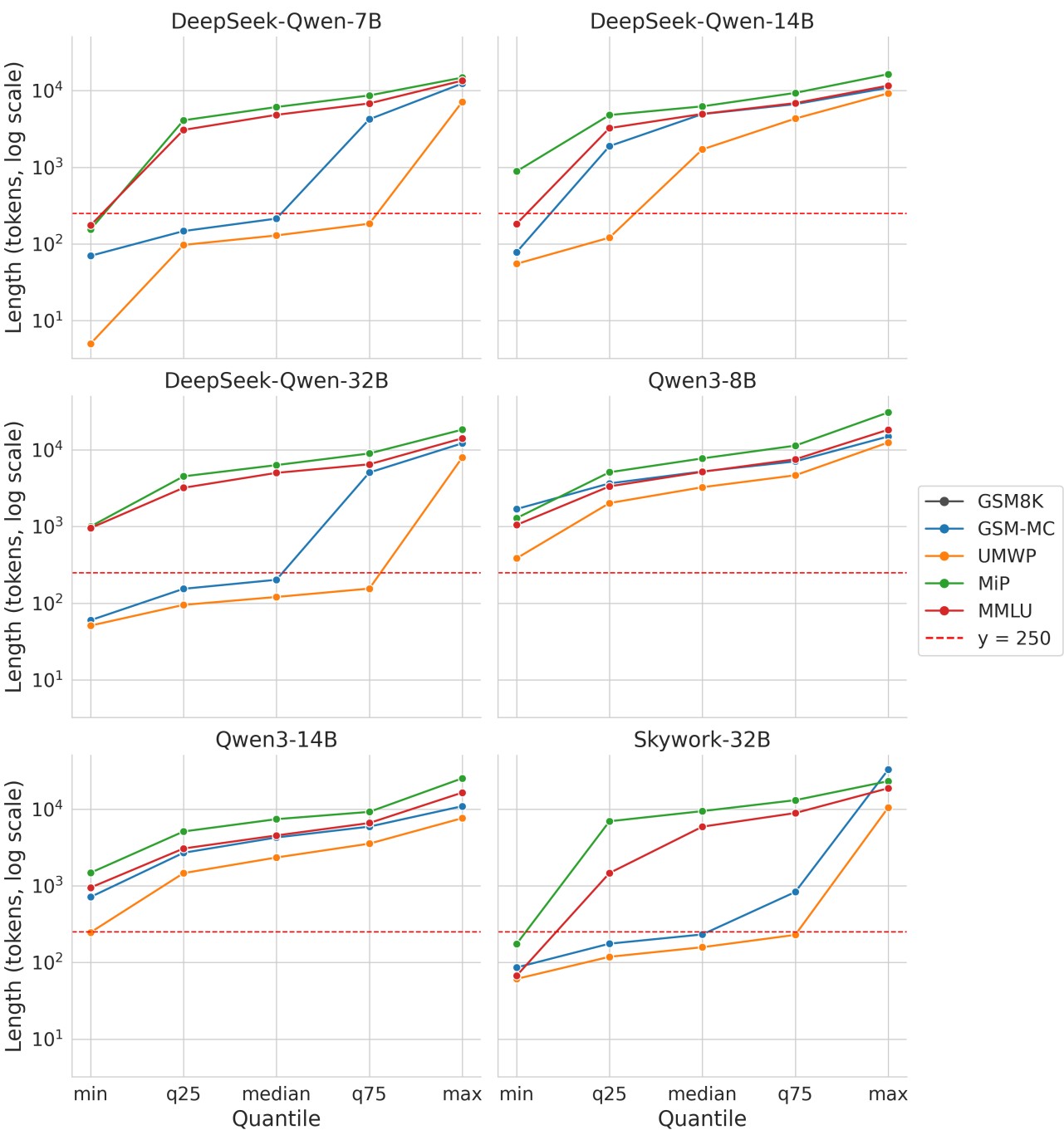

*Figure 12.* Quantiles of reasoning trace lengths (log scale) for UMWP for select DeepSeek and Skywork models. The red dashed line indicates $B = 250$, the interval at which uncertainty scores are evaluated. A large fraction of UMWP responses terminate before this point, limiting the opportunity for early stopping.

