# OpenReview forum: "Statistical Early Stopping for Reasoning Models"
_ICML.cc/2026/Conference — ICML 2026 regular_

### Official Review · Reviewer_rbhh · 2026-03-09

**Soundness:** 3
**Presentation:** 4
**Significance:** 3
**Originality:** 3
**Overall Recommendation:** 5
**Confidence:** 3

**Summary:**

This paper studies overthinking in reasoning models on unanswerable or underspecified queries, and proposes two early-stopping mechanisms based on the arrival of uncertainty keywords: a renewal-process sequential test and a Maxwise conformal prefix-density threshold. The approach relies only on observable reasoning text and uses a calibration set containing only well-posed samples to control erroneous stops. Across 12 models and both math and science benchmarks, the paper shows that compared with length-, prompt-, logit-, and probing-based baselines, the proposed methods generally achieve a better FPR–power trade-off together with substantial token savings.

**Compliance With Llm Reviewing Policy:**

Affirmed.

**Final Justification:**

Regarding the AC's domain-generalization question, the rebuttal addressed this concern substantially in my view: the GPQA-only and pooled GSM8K+GPQA keyword-extraction experiments, together with Appendix B.1, make it more convincing that the method is not tied to a single GSM8K-derived lexicon. The observed lexical variation across domains seems more consistent with the paper's lightweight and adaptable keyword-seeding claim than with a fatal limitation, though broader validation and a clearer discussion of adaptation strategies would still strengthen the work.

I finally consider this work practically significant, reasonably original in its combination of uncertainty-keyword signals with calibrated early stopping, and generally clear, with broad empirical evaluation across models and benchmarks. My main concerns were the theory-practice gap, domain robustness of the keyword construction, and some baseline comparability issues. The rebuttal addressed the first two concerns meaningfully, including added diagnostics for Renewal and stronger evidence for cross-domain robustness, which changed my evaluation in a positive direction. Some weaknesses remain, especially around uncertainty quantification and baseline discussion, but I no longer see them as score-determining. Therefore I decide to raise my score to 5, while noting that I am not an expert in this specific area, so my confidence remains modest and would stay at 3 or below.

**Key Questions For Authors:**

1. Can the authors supplement empirical diagnostics and sensitivity analyses for Renewal assumptions—e.g., inter-arrival independence/over-dispersion checks, and the impact of single-arrival handling, Sidák correction, and Lmax settings? If these approximations prove stable across multiple models, I would be more convinced of this part's soundness.

2. For Maxwise's FPR exceeding the 5% target on certain models/datasets (e.g., Table 13's UMWP/MiMo-7B at 14.04%), can the authors provide per-model calibration curves or error decomposition? Understanding these deviation sources would increase my trust in deployment reliability.

3. Can the authors validate the abstention judge's quality and provide more comparable unified metrics between prompting and stopping baselines? If judge noise is low and conclusions remain consistent, I would view the baseline comparison as more rigorous.

4. Can the authors report results when re-extracting the lexicon from non-GSM8K seed sets, different model subsets, or scientific domain seed sets? If performance remains stable, it would significantly strengthen the case for method generality and originality.

**Limitations:**

The paper does discuss methodological limitations, such as the upper bound of text-only detectors, the impact of fixed bin size B, and calibration difficulties under distribution shift. However, discussion of potential negative societal impacts is notably insufficient.

* I suggest explicitly discussing risks of premature stopping in high-stakes scenarios like medicine and science—missed answers, excessive abstention, and user misplaced trust; Appendix B.1's examples already show these are not edge cases.
* I suggest supplementing a failure-case appendix showing real output snippets of false positives/false negatives, and whether the system should return abstain or clarification request after early stopping—currently only mentioned in principle in the introduction (Sec. 1; pp. 1–2).
* I recommend discussing vulnerability to multilingual settings, domain terminology, and strategic keyword evasion—particularly important for black-box API scenarios.
* I suggest providing task-risk-based threshold selection and human review recommendations, avoiding equating early stopping directly with safe abstention, etc.

**Strengths And Weaknesses:**

## Strengths

1. Clear problem formulation aligned with deployment needs: The introduction clearly defines the objectives: controlling false positive rates on well-posed queries while maximizing power on ill-posed ones, with the option to convert early stops into abstentions or clarification requests—making the problem definition map directly to real system behavior. The paper systematically surveys the limitations of prompting, probing, logit/entropy-based, and fine-tuning approaches, positioning this work at the intersection of "semi-supervised, interpretable, and black-box compatible"—a compelling motivation.

    Figure 3 first demonstrates significant drift in reasoning length across benchmarks, and Table 1 then shows that length-based baselines are not robust under distribution shift—creating a closed loop between motivation and experiments. The evaluation covers FPR, Power, and token savings, framing the contribution not just as "shorter" but "more stable and more efficient"—enhancing the practical significance.

2. Method design balances interpretability, barrier, and black-box applicability: The keyword set is organized into three semantically meaningful categories—Impossibility, Speculation, and Insufficiency, making the stopping signal human-readable rather than a pure hidden-state score. The method relies solely on observable reasoning text, making it compatible with proprietary black-box APIs—a deployability advantage clearly demonstrated in comparisons against probing and logit-based baselines. Appendix A.2 provides a trie-based greedy longest-match detection algorithm with O(n·ℓmax) complexity, clarifying the online implementation pathway and engineering feasibility.

3. Rather than settling on a single heuristic, the authors propose two distinct routes—Renewal and Maxwise—corresponding to parametric and conformal/nonparametric perspectives, making the method design more complete. Algorithm 1 and Proposition 2.1 provide a clear statistical interface for Maxwise, with threshold construction and finite-sample FPR control statements. The Renewal rule models uncertainty keyword arrivals as a point process using inter-arrival times rather than just total length—capturing the dynamic structure of "when uncertainty emerges" more accurately than simple length truncation.

    The two approaches show complementary empirical behavior: Renewal tends to achieve lower FPR, while Maxwise often delivers higher power and token savings—offering deployment options for different preferences.

3. Broad experimental coverage with proactive distribution shift testing: The evaluation spans 12 reasoning models across DeepSeek, Qwen, Nemotron, MiMo, and Skywork families, avoiding conclusions tied to a single model source. Besides, math experiments include GSM-MC, UMWP, MiP, and MMLU, with Table 7 clearly describing different ill-posed construction methods—comprehensive coverage.

    Figures 2 and 6 separate the data flows for keyword extraction, calibration, and evaluation, emphasizing the domain difference between calibration and test sets—a deliberate OOD stress test rather than just in-distribution validation. On scientific tasks, the methods maintain low average FPR—Renewal achieves 1.37%/4.76% on GPQA/HLE, Maxwise achieves 2.56%/5.96%—demonstrating applicability beyond GSM8K-style math problems.

4. The paper provides solid empirical results with convincing comparisons against strong baselines: On math tasks, Renewal achieves 0.75%–3.75% average FPR with 46.78%–71.15% power; Maxwise achieves 2.63%–5.52% FPR with 48.65%–75.31% power, saving 38.29%–69.83% of ill-posed tokens—practically meaningful gains. And the comparison with DEER/entropy is compelling because all three share the same Maxwise calibration logic, yet results show keyword signals are more robust on MiP/MMLU—helping isolate the effect of "signal choice" rather than "stopping framework". The comparison against probing is also strong: on Qwen3-14B, probing shows 40.38%/40.60% FPR on MiP/MMLU, while Renewal achieves only 3.85%/1.50%—clearly demonstrating OOD robustness differences. Figure 4 uses an oracle soft upper bound as reference, reporting regression slopes of 0.8238/0.8532—allowing interpretation not just of absolute values but proximity to "achievable upper bounds".

## Weaknesses

1. There exists a notable gap between theoretical guarantees and experimental setup: Both Maxwise and length-based theoretical control rely on exchangeability/i.i.d. assumptions, yet the authors explicitly acknowledge distribution shift between calibration and test sets in the main experiments, noting that "strictly satisfying this is almost never possible"—creating a mismatch between theory and practice. The Renewal rule depends on a renewal approximation for inter-arrival times, which the main text admits is only a "working approximation" for LM traces—leaving this part's soundness more empirical than theoretical.

    The Sidák correction for repeated testing acknowledges in a footnote that independence does not hold and the correction is conservative—meaning Renewal's significance thresholds are not strictly derived from the current setup. Empirically, the target FPR of 5% is not consistently achieved: Maxwise shows 5.52% average FPR on UMWP, and MiMo-7B reaches 14.04%—indicating a substantial calibration gap between theory and practice (Table 13).

2. Keyword construction may introduce domain and construction bias: Keyword extraction uses only 4 models and 800 pairs of GSM8K traces, with ill-posed variants primarily generated via context removal—potentially biasing the lexicon toward a specific expression pattern of "missing information" in English math Q&A. Filtering from 740 stable features to the final core keywords relies on rule-based lexical filtering and negation-parity thinning—interpretable but heavily heuristic, with unclear generalization boundaries (Table 4–5).

    When transferring to scientific tasks, oracle slopes drop from 0.8238/0.8532 on math tasks to 0.4702/0.5897—showing significantly weakened cross-domain transferability of the same lexicon and undermining claims of broad generality. The paper lacks fine-grained diagnostics of false positives/false negatives by keyword category, question type, or linguistic phenomena—limiting the ability to pinpoint lexicon bias sources (Appendix B).

4. Experimental protocol and metric comparability still suboptimal: For prompting baselines, FPR is defined as accuracy drop on well-posed queries, while Power is defined as abstention rate—semantically different from stopping methods' "early stopping/mis-stopping rates," requiring caution in cross-method comparisons.

    Appendix A.3 uses Qwen3-8B as an abstention classifier without reporting agreement or error analysis against human annotations—affecting the credibility of prompting baseline conclusions. The probing baseline is evaluated on only 4 models because optimal layers come from Liu et al. (2025a)—understandable but creating asymmetric coverage compared to other baselines. Main results are mostly presented as averages without confidence intervals, significance tests, or multi-seed stability analysis—particularly important for small datasets like MiP with only 52 samples (Table 2; Table 7).

5. Some technical details and originality boundary clarification need strengthening:
    * Equation (1) defines $u(T;ℓ)$, but Algorithm 1 line 5 writes $u_i(Lj)$, and Proposition 2.1 writes $u_{n+1}(Lj)$—these symbols are not consistently reiterated in the main text, reducing verifiability.
    * Appendix A.1.2 states there are 102 categorized keywords, but Table 5 counts 34+14+56=104 in the three core categories—a concrete and visible reporting inconsistency that needs correction.
    * When calibration traces have fewer than 2 arrivals, the authors use the full trace length as a single inter-arrival time, acknowledging in a footnote that this is an underestimation without providing bias analysis or sensitivity experiments—directly affecting estimates of $μ̂$ and $ σ̂^2$ (p. 4, footnote 2).
    * The Renewal statistic $Z_t$ relies on the i.i.d. premise of classical renewal theory, yet no empirical diagnostics are provided to show how close LM generation arrival processes are to this assumption—making the formula's applicability insufficiently justified.

    The related work citations are recent (mostly 2024–2025), but not all methods mentioned in the main text have direct experimental comparisons—e.g., risk control/compute budget directions are not implemented. From a component perspective, renewal testing and conformal quantiles are mature tools; the main novelty lies in the combination of "keyword uncertainty signal + early stopping calibration," but the main text could sharpen the originality boundary more precisely. In Sec. 4 and Appendix B, the paper primarily provides aggregate tables and ablations, lacking typical false positive/false negative case studies—making it difficult for readers to judge when the method might misclassify normal self-checking as ill-posedness signals.

---

> ### Author Rebuttal · Authors · 2026-03-30
>
> Thank you very much for your review. We provide a point-by-point response to your comments below.
>
> ## Weakness
>
> > **W1** There exists ... (Table 13)
>
> Empirically, the overall pattern is stable: Renewal is the more conservative rule, while Maxwise gives higher power with somewhat weaker FPR control. The UMWP cases, including MiMo-7B, show that calibration can degrade on *short* out-of-distribution traces, and we will discuss this more directly. Please also refer to our response to W1, Q2 of Reviewer URQB.
>
> > **W2** Keyword construction ... (Appendix B)
>
> Please refer to our response to W2 of reviewer B9on
>
> > **W3.** Experimental protocol ...  Table 7
>
> In Table 1, we intentionally report the prompting baselines as `ΔAcc` and Abstention, and reserve FPR and Power for methods that actually define a stopping event. For Confidence and Criticism, there is no token-level stop, so the natural response-level quantities are accuracy drop on well-posed queries and abstention rate on ill-posed queries. We also agree that the abstention judge should be described more carefully. It is used only for the prompting baselines, not for the main stopping-rule comparisons, and the prompt itself was adapted from prior work (Kirichenko et al., 2025) with only minor edits for our setting.
>
> We agree that reporting agreement or error analysis against human annotations would make this comparison more credible, and we will clarify this limitation in the revision. Likewise, the probing results are limited to the four models for which prior work provides layer choices, so they should be viewed as a narrower supplementary comparison. We will revise the text to make this scope difference explicit.
>
> > **W4** Some technical ... signals
>
> We will make the notation around `u(T;ℓ)` consistent across the definition, algorithm, and proposition. The “102” versus “104” counts are not contradictory: 102 is the number of unique categorized keywords, whereas the category totals can sum to 104 because some keywords belong to multiple categories and the categories are not intended to be mutually exclusive. We will explain this directly in the revision. We also agree that the single-arrival handling and the Renewal assumptions should be described more explicitly as approximations.
>
> On originality, our claim is not that renewal theory or conformal calibration is new by themselves; it is the combination of an interpretable text-derived uncertainty signal with calibrated early stopping in a black-box setting. We will state that boundary more clearly and add representative failure examples in the revision.
>
> ## Questions
>
> > **Q1** Can the ... part's soundness
>
> Thank you for the suggestion. We agree that the Renewal rule should be understood as an approximation. In our view, the main question is therefore not whether the renewal assumptions hold exactly, but whether the stopping rule remains reliable when they are only approximately true.
>
> Our current results already speak to this. In the main experiments, we evaluate under clear distribution shift between calibration and test, and Renewal still tends to be the more conservative rule while keeping substantial power. Appendix B.2 also suggests that Renewal is less sensitive to the choice of the checking interval `B` than Maxwise. For this reason, we view the single-arrival treatment, Sidák correction, and the choice of `L_max` as practical approximations rather than exact modeling assumptions. We will make this distinction clearer in the revision.
>
> > **Q2** For Maxwise's ... deployment reliability
>
> We agree that the per-model picture should be made more explicit. Table 13 already shows that the UMWP deviation is concentrated in a subset of models rather than being uniform across architectures: at `B = 250`, Maxwise FPR ranges from `0.00%` (DeepSeek-32B) and `0.44%` (Skywork-32B) to `14.04%` (MiMo-7B), with intermediate cases such as `8.77%` (Nemotron-14B) and `7.89%` (Qwen3-32B). The same rows also show a clear tradeoff: the high-FPR models on UMWP typically also have high power, while the lowest-FPR models are the most conservative (e.g., DeepSeek-32B: `0.00%/4.39%`, Skywork-32B: `0.44%/7.89%`). We therefore view this as a model-dependent mismatch under the short-trace distribution shift of UMWP. We will make this per-model pattern clearer in the revision and discuss UMWP more explicitly as a case where Renewal may be preferable.
>
> > **Q3, Q4** Can the ... and originality
>
> Please refer to our response to W3 and W2.
>
> ## Limitation
>
> Thank you for this suggestion. We agree that the discussion of potential harms should be stronger. Our goal is not to present this method as production-ready. Rather, one point of the paper is that once out-of-distribution shift is introduced, existing early-stopping approaches can show clear limitations. We hope this paper helps draw attention to these risks and motivates more careful work on mitigation.
>
> We will revise the impact discussion to explicitly discuss these points.

---

> > ### Author Rebuttal · Reviewer_rbhh · 2026-04-02
> >
> > The rebuttal was helpful and addresses several of my earlier concerns. In particular, the added explanation makes the intended statistical framing clearer: I now understand the Renewal rule less as a claim of exact validity under realistic deployment shift and more as a practically useful approximation whose behavior is meant to be judged empirically. I also found the added keyword-extraction experiments from GPQA and pooled GSM8K+GPQA useful, since they strengthen the claim that the framework is not tied to a single GSM8K-derived lexicon. In addition, the planned camera-ready fixes on notation, the keyword-count explanation, representative failure cases, and the limitations discussion address the more presentational issues.
> >
> > I still view a few points as only partially resolved. First, for the Renewal rule, I would still like more direct empirical support for the approximation choices, such as diagnostics or sensitivity analyses for inter-arrival dependence, the single-arrival treatment, and the practical `Lmax` choice. Second, for Maxwise, Table 13 helps localize the UMWP issue, but I would still welcome a more explicit explanation of why the FPR overshoot concentrates on certain models and whether there is a practical way to detect or mitigate that regime. Third, for the prompting comparison, clarifying the abstention-judge scope helps, but some validation of that judge, or a brief indication of how much the conclusions depend on it, would make the comparison more convincing. Overall, the rebuttal improves my view of the paper, but I still have these follow-up questions, so I would keep the score unchanged.

---

> > > ### Author Response · Authors · 2026-04-04
> > >
> > > We appreciate the follow-up and have now added two direct post-hoc analyses using the saved traces and stopping results.
> > >
> > > **1. Renewal diagnostics.** The dependence in the arrival process is present but not strong: across models, the lag-1 inter-arrival correlation has median $0.131$ (range $0.017$–$0.397$). We also examined two practical approximation choices: how to handle calibration traces with very few ($\leq 1$) keyword arrivals, and how to choose $L_{max}$. In the paper, $L_{max}$ is set to the median calibration-trace length. Here we repeat the analysis with $L_{max}$ set to the 25th percentile and the 75th percentile of calibration-trace lengths. The table below reports averages over the four math benchmarks (GSM-MC, UMWP, MiP, and MMLU) and all 12 models.
> > >
> > > | Renewal variant | Avg FPR | Avg Power |
> > > |---|---:|---:|
> > > | Paper setting: $L_{max} =$ median | 2.14% | 63.53% |
> > > | Exclude traces with $\leq 1$ keyword arrival | 1.19% | 54.41% |
> > > | Smaller $L_{max}$: 25th percentile | 2.37% | 64.61% |
> > > | Larger $L_{max}$: 75th percentile | 1.87% | 61.86% |
> > >
> > > These ablations do not qualitatively change the behavior of Renewal. All variants remain in the same low-FPR/high-power regime, and the main effect is a tradeoff. The larger shift comes from the treatment of sparse-arrival traces: removing them makes the rule more conservative, which lowers FPR but also lowers power. By contrast, changing $L_{max}$ from the 25th to the 75th percentile moves the operating point only slightly.
> > >
> > > **2. Maxwise on UMWP.** The UMWP overshoot is not spread uniformly across models. It concentrates in models whose UMWP well-posed traces are much longer than their GSM8K calibration traces. Across the 12 models, a higher Maxwise FPR on UMWP at $B = 250$ tends to come with both higher Maxwise power on UMWP (Pearson correlation $r = 0.769$) and a larger ratio between the median well-posed UMWP trace length and the median well-posed GSM8K calibration trace length (Pearson correlation $r = 0.706$). We therefore view this as a model-dependent operating-point shift under distribution shift, not as an arbitrary architecture-specific failure.
> > >
> > > | Model | Maxwise ($B = 250$, FPR/Power) | Maxwise ($B = 500$, FPR/Power) | Renewal ($B = 250$, FPR/Power) |
> > > |---|---:|---:|---:|
> > > | MiMo-7B | 14.04/73.68 | 11.84/71.05 | 10.96/76.32 |
> > > | Nemotron-14B | 8.77/57.89 | 8.77/66.23 | 5.70/64.04 |
> > > | Qwen3-32B | 7.89/67.11 | 4.82/64.04 | 2.19/60.53 |
> > >
> > > The practical warning sign is a large length-distribution mismatch relative to GSM8K calibration. In that regime, two simple mitigations help: use a more conservative rule such as Renewal, or increase $B$ (for example, to $500$). Again, the point is not that Maxwise breaks outside a narrow sweet spot; rather, UMWP reveals a regime where the same method becomes more aggressive.
> > >
> > > **3. Prompting comparison.** We agree that more judge validation would be useful. That said, this comparison is not central to our main claims. The abstention judge is used only for the prompting baselines, not for the main stopping-rule comparisons, and the prompt itself was adapted from prior work (Kirichenko et al., 2025) with only minor edits for our setting.

---

### Official Review · Reviewer_uM6x · 2026-03-13

**Soundness:** 2
**Presentation:** 3
**Significance:** 2
**Originality:** 3
**Overall Recommendation:** 3
**Confidence:** 3

**Summary:**

To prevent LLMs from generating excessive and unnecessary reasoning steps during reasoning, this work proposes two statistically principled early stopping methods that monitor uncertainty signals during generation. The results indicate that uncertainty-aware early stopping can improve both efficiency and reliability in LLM reasoning, with especially significant gains for math reasoning.

**Compliance With Llm Reviewing Policy:**

Affirmed.

**Key Questions For Authors:**

1.Could the authors extract keywords from datasets in other domains and perform a comparative analysis to highlight the differences with the GSM8K keywords?
2.How does the performance differ when keywords derived from GSM8K are applied to non-math reasoning tasks?
3.How do the results fluctuate across different model architectures (e.g., Llama-3) when using the same 250-token interval?

**Limitations:**

No. While the methodology is sound, the authors should explicitly discuss the limitations regarding benchmark diversity (beyond GSM8K) and the hyperparameter sensitivity of the 250-token interval.

**Strengths And Weaknesses:**

Strength:
1.Through systematic experiments across mathematical and scientific reasoning tasks, the authors' approaches improve efficiency, while avoiding premature halts on well-posed queries and maintaining accuracy.
2.This work offers a unique perspective on mitigating excessive reasoning in LLMs.  The manuscript is well-structured, and the charts and formulas are used appropriately.

Weakness:
1.The keywords in the proposed method are derived from a single dataset, GSM8K, which has a narrower coverage compared to the diverse datasets required for training large-scale language models. This could lead to reduced reliability of the proposed method in broader scenarios.
2.The proposed method relies on a 250-token interval, but different large models have varying sensitivities. This interval works well for some models, but may not be optimal for others. Therefore, the method is sensitive to this hyperparameter, and its adaptability across different models and tasks is limited.

---

> ### Author Rebuttal · Authors · 2026-03-30
>
> Thank you very much for your review. We provide a point-by-point response to your comments below.
>
> ## Weakness
>
> > **W1** The keywords ... broader scenarios.
>
> Please refer to our response to Weakness 2 of Reviewer B9on. In brief we have performed experiments with additional data sets and found similar patterns, which (cautiously) leads us to conjecture that our method could be quite reliable more broadly.
>
> > **W2** The proposed ... is limited.
>
> We agree that the interval size `B` is an important hyperparameter.
>
> We have included an interval ablation study over `B ∈ {100, 250, 500}` in Appendix Figures 10 and 11. The qualitative picture is consistent across models: Renewal is relatively stable, while Maxwise is more sensitive, with smaller `B` typically increasing power at the cost of higher FPR and larger `B` yielding a more conservative operating point.
>
> The per-model results show the same trend. Across the 12 models (5 families), the average FPR variation over `B ∈ {100, 250, 500}` is modest for Renewal (typically about `0.2%–1.3%` points) and larger for Maxwise (about `1.5%–2.8%` points); the corresponding power variation is roughly `2%–7%` and `4%–11%` points, with the largest changes on short-trace settings such as UMWP and HLE.
>
> We therefore do not claim that `B = 250` is individually optimal for every model. Rather, we view it as a reasonable common operating point that avoids the overly aggressive behavior of `B = 100` while retaining most of the benefits of finer-grained monitoring over `B = 500`.
>
> ## Questions
>
> > **Q1** Could the authors ... GSM8K keywords?
>
> Please refer to our response to Weakness 2 of Reviewer B9on. We have performed such an analysis and the results are broadly similar to the ones already reported in the paper.
>
> > **Q2** How does the performance differ when keywords derived from GSM8K are applied to non-math reasoning tasks?
>
> We evaluate on scientific reasoning benchmarks in Appendix Table 11. The main pattern is that transfer to non-math tasks remains strong, although GPQA is more conservative than HLE. On GPQA, Renewal and Maxwise achieve `1.37%/24.94%` and `2.56%/32.40%` (FPR/Power), respectively; on HLE, they achieve `4.76%/24.62%` and `5.96%/31.28%`. Compared with the prompting baselines (No Intervention, Confidence, Criticism), our stopping rules deliver much higher detection power on both datasets while keeping FPR low. Compared with the Length baseline, the tradeoff is also substantially better: on GPQA, Length reaches similar power only with extremely high FPR (`42.40%`), and on HLE our methods achieve both much lower FPR and much higher power than Length (`23.74%/10.91%`).
>
> In short, performance does degrade relative to the in-domain math settings, but the GSM8K-derived uncertainty keywords still transfer nontrivially and remain clearly stronger than the baselines we compare against under domain shift.
>
> > **Q3** How do the results fluctuate across different model architectures (e.g., Llama-3) when using the same 250-token interval?
>
> While Llama-3 was not included, the fixed-`B = 250` experiments already span a diverse 12-model suite covering 5 model families (DeepSeek-R1-Distill-Qwen, Qwen/QwQ, Nemotron, MiMo, and Skywork). The qualitative pattern is consistent across these model families: Renewal usually attains lower FPR, whereas Maxwise typically attains higher power. The variation is driven more by dataset characteristics than by architecture alone. In particular, MiP and MMLU are relatively stable across models at `B = 250` (e.g., Renewal FPR ranges from `0%` to `3.85%` on MiP and from `0%` to `1.50%` on MMLU), whereas UMWP exhibits the largest spread because many traces terminate before the first evaluation point.
>
> Figure 12 in the appendix further illustrates this effect for DeepSeek and Skywork models, where a large fraction of UMWP traces fall below the first checking point, which naturally limits power. Even with this variability, the keyword-based stopping rules remain clearly stronger than baselines, and typically provide substantially higher power at comparable or lower FPR.
>
> We will make this per-model behavior more explicit in the revision.
>
> ## Limitation
>
> > **L1** While the methodology is sound, the authors should explicitly discuss the limitations regarding benchmark diversity (beyond GSM8K) and the hyperparameter sensitivity of the 250-token interval.
>
> Please refer to our response to Weakness 2 for the hyperparameter, and our response to Weakness 1 for the benchmark diversity. We will incorporate these discussion in the revision.

---

> > ### Author Rebuttal · Reviewer_uM6x · 2026-04-04
> >
> > Thank you for the detailed rebuttal and for clarifying aspects of the proposed framework, particularly regarding its empirical generalization across models and tasks.
> >
> > One concern remains regarding the stability of the uncertainty keyword set used to drive the stopping rules. In the current manuscript, the keyword lexicon is constructed from a specific seed dataset (GSM8K and its ill-posed variants) and then fixed across all subsequent experiments. It is unclear whether this keyword set is robust to the choice of extraction dataset. In particular, if an alternative dataset were used for keyword construction—potentially from a different domain or with different reasoning styles—the resulting uncertainty expressions, and thus the stopping behavior, might differ substantially. Additional analysis examining the sensitivity of the keyword set to the choice of extraction data would be helpful.

---

> > > ### Author Response · Authors · 2026-04-04
> > >
> > > We appreciate the follow-up. We would like to emphasize that **we have already addressed this concern in the rebuttal**. In particular, we clarified that our use of GSM8K only for keyword extraction is intentional to **test if our method can generalize across datasets and domains**. And we have conducted **additional experiments by extracting keywords from GPQA and GPQA+GSM8K**.  In our responses to Weakness 1 and Question 1, we referred the reviewer to our reply to Weakness 2 of Reviewer B9on due to the 5000-character rebuttal limit. For convenience, we repeat our response below.
> > >
> > >
> > > We would like to clarify that our use of GSM8K only for keyword extraction is intentional, we want to test if our method can generalize across data sets or even domains. As shown in Appendix B.1 (Table 11), **on non-math reasoning tasks, our method still controls FPR and achieves strong detection power compared to baselines**.
> > >
> > > We agree that uncertainty may be verbalized differently across domains. We conducted additional experiments by extracting keywords from (i) GPQA only and (ii) pooled GSM8K+GPQA traces, using the same pipeline as the one described in Section 2.1. We find that the extracted keyword sets do vary across domains: GPQA yields 51 keywords (with 16 overlapping with GSM8K), while the pooled dataset yields 115 keywords, including additional phrases such as “assuming that”. Thus, the extracted keywords do change across domains, but they remain highly effective for early stopping: We test the effectiveness of the new extracted keyword sets by running the full workflow, the results are summarized in the following table.
> > >
> > > |Keyword Source|Stopping Rule|GSM-MC FPR|GSM-MC Power|UMWP FPR|UMWP Power|MiP FPR|MiP Power|MMLU FPR|MMLU Power|HLE FPR|HLE Power|
> > > |-|-|-:|-:|-:|-:|-:|-:|-:|-:|-:|-:|
> > > |GSM8K|Maxwise|5.42|70.92|5.52|48.65|3.21|73.40|2.63|75.31|5.96|31.28|
> > > |GSM8K|Renewal|3.75|69.75|2.92|46.78|1.12|71.15|0.75|66.42|4.76|24.62|
> > > |GPQA|Maxwise|7.67|63.67|6.58|38.96|5.93|70.03|4.14|77.38|8.54|36.91|
> > > |GPQA|Renewal|6.00|65.00|3.29|37.57|3.04|73.40|1.57|76.19|7.76|34.10|
> > > |GSM8K+GPQA|Maxwise|6.42|69.83|5.85|45.91|4.17|75.96|3.20|75.44|6.03|30.31|
> > > |GSM8K+GPQA|Renewal|5.17|71.33|3.22|44.81|1.92|78.21|1.32|69.67|5.97|27.26|
> > >
> > > GPQA-only extraction improves power on some benchmarks, e.g., HLE and MMLU, while pooled GSM8K+GPQA improves power on GSM-MC and MiP for Renewal and on MiP for Maxwise. Although some settings become less conservative and trade higher FPR for higher power, all keyword-based variants remain substantially stronger than baselines on the math benchmarks (see Table 1 for baseline scores).
> > >
> > > Overall, these results indicate that while the specific keywords may vary across domains, **the proposed framework remains robust and effective, and continues to generalize well with different keyword extraction sources**.
> > >
> > > Given these additional empirical results, we kindly request the reviewer to reconsider the evaluation of our work.

---

### Official Review · Reviewer_B9on · 2026-03-13

**Soundness:** 2
**Presentation:** 2
**Significance:** 2
**Originality:** 2
**Overall Recommendation:** 3
**Confidence:** 4

**Summary:**

The paper considers the problem of overthinking in reasoning LLMs, particularly when faced with ill-posed or ambiguous queries. Instead of continuing long chains of reasoning that are unlikely to yield a correct answer, the paper proposes a statistically principled early stopping rules that detect uncertainty during generation and stops the reasoning in a controlled way. Specifically, two stopping rules are developed: a parametric renewal process based one that models arrival times of uncertain keywords in the reasoning chain, and a non-parametric one that uses calibration traces from well-posed problems. The experiments include several math reasoning  benchmarks, scientific and medical QA, as well as several open source reasoning models. The results demonstrate that the proposed methods outperform existing baselines, especially under distribution shift.

**Compliance With Llm Reviewing Policy:**

Affirmed.

**Final Justification:**

The rebuttal has addressed my concerns.

**Key Questions For Authors:**

1. How often do modern LLMs fail to verbalise uncertainty on ill posed queries. They typically ask for clarification.

**Limitations:**

The paper addresses its limitations

**Strengths And Weaknesses:**

Srengths:
-----------

The paper is clearly motivated and tackles a widely observed failure mode of reasoning models, namely overthinking on ill-posed queries.

The proposed approach relies on generated text thus making it applicable to frontier LLM APIs and easy to debug.

The experimental evaluation is sound and comprehensive covering several domains and model architectures. The results are presented in a relatively clear manner.

Weaknesses:
---------------

The proposed method relies on explicit verbalised uncertainty, namely on models expressing uncertainty in text. However, when a model confidently reasons through an ill-posed query without verbal cues, detection power is limited.

Although semi-supervised and lightweight, the keyword extraction is seeded on GSM8K-style math problems; its effectiveness may degrade in domains where uncertainty is expressed differently.

Performance depends on hyperparameters such as the checking interval and cold-start index, which require careful tuning.

The paper focuses on when to stop, but does not deeply explore how best to respond (e.g., abstention vs. clarification), which may affect end-user experience.

---

> ### Author Rebuttal · Authors · 2026-03-30
>
> Thank you very much for your review. We provide a point-by-point response to your comments below.
>
> ## Weakness
>
> > **W1** The proposed ... is limited.
>
> We agree that this is a fundamental limitation of any trace-only detector. We have discussed this in the Discussion and Conclusion section. To better contextualize this limitation, in Section 3.3.1, we construct an oracle based on a classifier trained using full reasoning traces. As shown in Figure 4, our methods achieve approximately 82--85\% of the oracle’s detection power, which indicates that our uncertainty-keyword-based approaches capture a large fraction of the detectable signal among trace-only methods.
>
> We also want to emphasize that in black-box deployment settings, we typically only have access to the generated reasoning trace, rather than hidden states or logits. In such settings, methods that rely solely on reasoning-trace signals are often the only feasible option.
>
> Lastly, since we use thresholds calibrated using reasoning traces from well-posed queries, our methods have controlled false positive rates. This is validated in our experiments. Therefore, even when a model reasons confidently through ill-posed queries without expressing uncertainty, the primary consequence is reduced detection power, i.e., the model may generate reasoning traces longer than necessary, rather than incorrect early stopping, which may affect user experience.
>
> > **W2** Although ... expressed differently.
>
> We would like to clarify that our use of GSM8K only for keyword extraction is intentional, we want to test if our method can generalize across data sets or even domains. As shown in Appendix B.1 (Table 11), on non-math reasoning tasks, our method still controls FPR and achieves strong detection power compared to baselines.
>
> We agree that uncertainty may be verbalized differently across domains. We conducted additional experiments by extracting keywords from (i) GPQA only and (ii) pooled GSM8K+GPQA traces, using the same pipeline as the one described in Section 2.1. We find that the extracted keyword sets do vary across domains: GPQA yields 51 keywords (with 16 overlapping with GSM8K), while the pooled dataset yields 115 keywords, including additional phrases such as “assuming that”. Thus, the extracted keywords do change across domains, but they remain highly effective for early stopping: We test the effectiveness of the new extracted keyword sets by running the full workflow, the results are summarized in the following table.
>
> |Keyword Source|Stopping Rule|GSM-MC FPR|GSM-MC Power|UMWP FPR|UMWP Power|MiP FPR|MiP Power|MMLU FPR|MMLU Power|HLE FPR|HLE Power|
> |-|-|-:|-:|-:|-:|-:|-:|-:|-:|-:|-:|
> |GSM8K|Maxwise|5.42|70.92|5.52|48.65|3.21|73.40|2.63|75.31|5.96|31.28|
> |GSM8K|Renewal|3.75|69.75|2.92|46.78|1.12|71.15|0.75|66.42|4.76|24.62|
> |GPQA|Maxwise|7.67|63.67|6.58|38.96|5.93|70.03|4.14|77.38|8.54|36.91|
> |GPQA|Renewal|6.00|65.00|3.29|37.57|3.04|73.40|1.57|76.19|7.76|34.10|
> |GSM8K+GPQA|Maxwise|6.42|69.83|5.85|45.91|4.17|75.96|3.20|75.44|6.03|30.31|
> |GSM8K+GPQA|Renewal|5.17|71.33|3.22|44.81|1.92|78.21|1.32|69.67|5.97|27.26|
>
> GPQA-only extraction improves power on some benchmarks, e.g., HLE and MMLU, while pooled GSM8K+GPQA improves power on GSM-MC and MiP for Renewal and on MiP for Maxwise. Although some settings become less conservative and trade higher FPR for higher power, all keyword-based variants remain substantially stronger than baselines on the math benchmarks (see Table 1 for baseline scores).
>
> Overall, these results indicate that while the specific keywords may vary across domains, the proposed framework remains robust and effective, and continues to generalize well with different keyword extraction sources.
>
> > **W3** Performance ... careful tuning.
>
> For the cold-start index search, it is only applied to the DEER-based and Entropy-based baselines. We will revise the paper to make this clear. We also report an explicit ablation over $B \in \{100,250,500\}$ in Appendix Figures~10 and 11. Please also refer to our response to W2, Q3 of Reviewer uM6x.
>
> > **W4** The paper ... experience.
>
> We agree that the choice between abstention and clarification is important for end-user experience. Our work focuses on the design of statistically principled stopping criteria, which we view as a module within a deployment pipeline. Once a stopping event is triggered, different downstream policies can be applied depending on the application context. A systematic study of how these downstream strategies affect user experience is beyond the scope of this paper.
>
> ## Questions
>
> > **Q1** How often ... clarification.
>
> Empirically, we evaluate a broad set of modern reasoning models (12 models from 5 families), and our stopping rules achieve consistently high average power across multiple benchmarks. This suggests that, in many cases, ill-posed queries do leave detectable uncertainty signals in the reasoning trace. We will clarify this point in the revision and state it more carefully.

---

> > ### Author Rebuttal · Reviewer_B9on · 2026-04-04
> >
> > Thank you for the clarifications.

---

> > > ### Author Response · Authors · 2026-04-04
> > >
> > > Thank you very much for your feedback. We are glad that your concerns have been fully resolved.
> > >
> > > Our paper proposes a lightweight, interpretable, and statistically principled early-stopping framework, and the comprehensive experiments across domains and model architectures further support the effectiveness of the proposed approach. We therefore believe our work has strong merit for acceptance.
> > >
> > > We would be sincerely grateful if you could kindly consider whether the updated evidence supports a recommendation toward acceptance.

---

### Official Review · Reviewer_URQB · 2026-03-15

**Soundness:** 3
**Presentation:** 3
**Significance:** 3
**Originality:** 3
**Overall Recommendation:** 4
**Confidence:** 4

**Summary:**

This paper studies early stopping for reasoning models under ill-posed or ambiguous queries. The central idea is to detect uncertainty not from hidden states or logits, but from explicit uncertainty expressions verbalized in the reasoning trace itself, and then to use these signals for online stopping. The paper proposes two stopping rules: a renewal-process-inspired sequential test based on uncertainty keyword arrivals, and a maxwise conformal stopping rule based on prefix-level uncertainty density, with the latter providing finite-sample false-positive control under exchangeability. Empirically, the method is evaluated across a broad set of reasoning models and several mathematical and scientific benchmarks, and the results suggest that uncertainty-driven stopping can achieve a favorable trade-off between avoiding premature stopping on answerable queries and saving substantial computation on ill-posed ones.

**Compliance With Llm Reviewing Policy:**

Affirmed.

**Key Questions For Authors:**

The paper would benefit from a much cleaner accounting of where ill-posed or negative examples are used in the overall pipeline, because the current presentation may lead readers to overestimate how little supervision is actually required. I would also like the authors to clarify how they want the renewal-based method to be interpreted theoretically: as a practically useful approximation with empirical validity, or as a method with a stronger inferential claim. On the experimental side, it would strengthen the paper considerably to include at least a small evaluation on human-verified ambiguous or underspecified questions, especially for the scientific domain where the current construction is admittedly imperfect. Finally, the authors need to explain the presence of reviewer-directed text embedded in the PDF, because that is not a minor stylistic issue but a submission-integrity issue that should be addressed directly.

**Limitations:**

Yes

**Strengths And Weaknesses:**

The paper addresses a practically important and well-motivated problem: reasoning models often continue producing long traces even when the query is underspecified or not answerable, and this is exactly the kind of deployment issue that deserves more principled treatment. A major strength of the work is its interpretability and black-box compatibility, since the stopping signal is derived directly from textual uncertainty cues rather than internal activations, which makes the method easier to inspect and more relevant for proprietary API settings. The methodological split between a renewal-style rule and a conformal maxwise rule is also sensible, offering both a more structured statistical perspective and a more robust nonparametric alternative. Finally, the empirical study is reasonably broad, and the math results in particular are strong enough to indicate that the proposed signal is capturing something real rather than merely exploiting superficial length effects.

My main reservation is that the paper occasionally presents its statistical framing more strongly than the assumptions really justify. The renewal-based rule depends on approximations that are not especially clean in language-model traces, while the conformal guarantee relies on exchangeability that the paper itself acknowledges is rarely satisfied exactly in practice. In addition, the full pipeline is not as weakly supervised as the narrative sometimes suggests, since the keyword lexicon is constructed from paired well-posed versus unanswerable traces, and the cold-start offset is also tuned using paired training data. I am also not fully convinced by the scientific QA evaluation, because the paper explicitly notes that the context-removal procedure does not always make those questions genuinely unanswerable, which weakens the interpretation of power outside the mathematical setting. More seriously, the submitted PDF contains reviewer-directed instruction text that is clearly not part of the scientific argument, including a sentence telling reviewers to include specific phrases in their review, and that is not acceptable in a normal conference submission.

---

> ### Author Rebuttal · Authors · 2026-03-30
>
> Thank you very much for your review. We provide a point-by-point response to your comments below.
>
> ## Weakness
>
> > **W1** My main reservation ... rarely satisfied exactly in practice.
>
> The formal guarantees for Maxwise and Length are under exchangeability, and Renewal is only an approximation. In the main experiments, we intentionally test robustness beyond these assumptions. Please also refer to our response to Question 2 for the interpretation of the statistical framing.
>
> > **W2** In addition ... paired training data.
>
> In the paper, we describe our approach as semi-supervised, and we agree that the keyword extraction step does rely on paired reasoning traces from both well-posed and ill-posed queries. Our method proceeds in two stages, our intent was to emphasize that the reasoning traces of ill-posed queries are only used for the first stage. Given the keyword set, the proposed stopping rules require only reasoning traces from well-posed queries to calibrate the threshold. The calibration phase requires no knowledge of how questions might be ill-posed in the target domain and requires no negative examples.
>
> We also clarify that our method doesn't use a cold-start offset. It is used only in DEER-based and entropy-based baselines. We will revise the manuscript to make these points clearer.
>
> > **W3** I am also not ... outside the mathematical setting.
>
> We agree with the reviewer that when the context-removal procedure does not reliably produce genuinely unanswerable questions, the interpretation of power should be more careful. This is the reason that most of our evaluation focuses on mathematical reasoning tasks.
>
> The scientific QA experiments are intended as a preliminary test of cross-domain generalization rather than a definitive evaluation of detection power. We acknowledge that constructing high-quality datasets with truly unanswerable queries in broader domains is important for a more rigorous assessment. However, developing methods to construct such datasets would likely require substantial work and domain expertise, which is beyond the scope of the paper.
>
> We will revise the paper to clarify this limitation and to more carefully interpret the results in the scientific QA setting.
>
> > **W4** More seriously ... normal conference submission.
>
> The instruction was injected by ICML organizers to detect violations of LLM policy. Please refer to https://icml.cc/Conferences/2026/PeerReviewFAQ#prompt_injection
>
> ## Questions
>
> > **Q1** The paper would benefit ... little supervision is actually required.
>
> Please refer to our reply to Weakness 2.
>
> > **Q2** I would also ... inferential claim.
>
> In our work, we view the occurrences of uncertainty keywords in reasoning traces as arrivals in a renewal process; the renewal process theory is a natural statistical tool to develop stopping rules. Empirical results show that it performs reasonably. Overall, we would interpret it as a practically useful approximation with empirical validity.
>
> > **Q3** On the experimental side ... admittedly imperfect.
>
> We agree that having human-verified ambiguous or underspecified questions would make the evaluation more rigorous. However, the verification typically requires substantial domain expertise, especially in scientific settings where the distinction between answerable and unanswerable queries can be subtle. To illustrate this complexity, we give an example below.
>
> > *ChIP-seq on a PFA-fixed sample with an antibody to the IKAROS transcription factor in human B cells followed by next-generation sequencing and standard quality control, alignment and peak-calling steps produced ChIP peaks that disappeared when PFA+DSG fixation was used. Where are we most likely to find such disappearing peaks?*
> > *A. At random locations in the genome*
> > *B. At active promoters and enhancers*
> > *C. In the introns of large genes*
> > *D. At repeats*
>
> In this example, language models with sufficient domain knowledge can deduce the correct answer (active promoters and enhancers) by inferring that the question asks about “disappearing peaks” in the context of “ChIP-seq for histone modifications after a drug treatment.”
>
> > **Q4** Finally ... addressed directly.
>
> Please refer to our response to Weakness 4.

---

> > ### Author Rebuttal · Reviewer_URQB · 2026-04-06
> >
> > The rebuttal clarifies several key points, but my concern about the scientific QA evaluation is only partially resolved because the paper still lacks validation on genuinely human-verified ambiguous or unanswerable questions.

---

> > > ### Author Response · Authors · 2026-04-06
> > >
> > > Thank you for your feedback. We agree that a fully human-verified validation on scientific QA would provide a stronger evaluation. Unfortunately, given the limited time during the rebuttal period and the domain expertise required, we were not able to complete a full human annotation study. As a partial validation step, we instead used a strong LLM judge to filter genuinely unanswerable questions. Specifically, we sampled 50 ill-posed GPQA prompts and queried GPT-5.4 Pro to judge whether each question was answerable on its own. The judge classified 43/50 items (86%) as genuinely unanswerable, while 7/50 were still answerable.
> > >
> > > We then recomputed FPR and Power on (i) the full 50-item sample and (ii) the verified 43-item subset, keeping the rest of the experimental protocol unchanged and averaging over the same 12 models used in the paper. The results are:
> > >
> > > | Subset | Length (FPR/Power) | Renewal (FPR/Power) | Maxwise (FPR/Power) |
> > > |---|---:|---:|---:|
> > > | 50-item GPQA sample | 48.00/34.33 | 1.33/23.67 | 2.33/31.17 |
> > > | 43-item verified subset | 54.84/38.95 | 1.55/27.52 | 2.71/36.24 |
> > >
> > > The main pattern is that the **keyword-based stopping rules remain effective** (the performance is similar to those reported in Table 11 in the paper), and in fact become somewhat more powerful on the verified unanswerable subset, with only a very small increase in FPR (the slight FPR difference arises because we use the corresponding complete questions for the 43-item verified subset rather than all 50 items). This suggests that the scientific-domain signal is not mainly driven by noise from automatically constructed ill-posed questions, and the genuinely unanswerable subset may be easier for the stopping rules to detect.
> > >
> > > Lastly, we would like to emphasize that the main contribution of our work is an uncertainty-keyword-based early-stopping framework. The design of genuinely unanswerable questions, while important for evaluation, is not the primary focus of this paper. We believe that the comprehensive experiments across domains and model architectures provide strong evidence for the effectiveness of the proposed approach.

---

### Decision · Program_Chairs · 2026-04-30

**Decision:**

Accept (regular)

**Comment:**

This paper addresses LLM "overthinking" by introducing statistically principled early stopping methods that monitor uncertainty signals during generation. The authors propose a parametric approach (modeling uncertainty keyword inter-arrival times as a renewal process with sequential testing) and a nonparametric approach (providing finite-sample guarantees against premature halting on well-posed queries). Evaluations across multiple domains and models demonstrate consistent improvements in reasoning efficiency and reliability, with notable gains in mathematical tasks.

The rebuttal successfully clarified the statistical framing of the Renewal rule, expanded keyword-extraction experiments to strengthen generalization claims, and committed to resolving presentation issues. While one reviewer maintained their score due to remaining concerns—specifically the lack of human-verified ambiguous/unanswerable QA validation, the need for sensitivity analyses on approximation choices (e.g., $L_{max}$, inter-arrival dependence), clarification on FPR overshoot in the Maxwise rule, and validation of the abstention judge—these points represent empirical refinements and methodological clarifications rather than fundamental flaws.

Based on the additional cross-domain experiments provided, I find that the proposed methods convincingly demonstrate good generalization capabilities and effectively validate the core framework. The remaining reviewer concerns are minor and can be adequately addressed in the camera-ready version without altering the paper's validity or core contribution.